# Achieving low-power single-wavelength-pair nanoscopy with NIR-II continuous-wave laser for multi-chromatic probes

Xin Guo [1,4], Rui Pu[1,4], Zhimin Zhu[1], Shuqian Qiao[1], Yusen Liang[1], Bingru Huang[1], Haichun Liu [2], Lucía Labrador-Páez [2], Uliana Kostiv[2], Pu Zhao[1], Qiusheng Wu[1], Jerker Widengren [2] & Qiuqiang Zhan [1,3✉]

Stimulated emission depletion (STED) microscopy is a powerful diffraction-unlimited technique for fluorescence imaging. Despite its rapid evolution, STED fundamentally suffers from high-intensity light illumination, sophisticated probe-defined laser schemes, and limited photon budget of the probes. Here, we demonstrate a versatile strategy, stimulated-emission induced excitation depletion (STExD), to deplete the emission of multi-chromatic probes using a single pair of low-power, near-infrared (NIR), continuous-wave (CW) lasers with fixed wavelengths. With the effect of cascade amplified depletion in lanthanide upconversion systems, we achieve emission inhibition for a wide range of emitters (e.g., $Nd^{3+}$, $Yb^{3+}$, $Er^{3+}$, $Ho^{3+}$, $Pr^{3+}$, $Eu^{3+}$, $Tm^{3+}$, $Gd^{3+}$, and $Tb^{3+}$) by manipulating their common sensitizer, i.e., $Nd^{3+}$ ions, using a 1064-nm laser. With $NaYF_4$:Nd nanoparticles, we demonstrate an ultrahigh depletion efficiency of 99.3 ± 0.3% for the 450 nm emission with a low saturation intensity of 23.8 ± 0.4 kW cm$^{-2}$. We further demonstrate nanoscopic imaging with a series of multi-chromatic nanoprobes with a lateral resolution down to 34 nm, two-color STExD imaging, and subcellular imaging of the immunolabelled actin filaments. The strategy expounded here promotes single wavelength-pair nanoscopy for multi-chromatic probes and for multi-color imaging under low-intensity-level NIR-II CW laser depletion.

[1] Centre for Optical and Electromagnetic Research, Guangdong Provincial Key Laboratory of Optical Information Materials and Technology, National Center for International Research on Green Optoelectronics, South China Academy of Advanced Optoelectronics, South China Normal University, Guangzhou 510006, P. R. China. [2] Experimental Biomolecular Physics, Department of Applied Physics, KTH Royal Institute of Technology, SE-106 91, Stockholm, Sweden. [3] MOE Key Laboratory and Guangdong Provincial Key Laboratory of Laser Life Science, College of Biophotonics, South China Normal University, Guangzhou 510631, P. R. China. [4] These authors contributed equally: Xin Guo, Rui Pu. ✉email: zhanqiuqiang@m.scnu.edu.cn

Super-resolution fluorescence microscopy encompasses a variety of techniques that can break the diffraction limit, and are of utmost importance for studies in life science and beyond[1–7]. As a typical super-resolution technique, stimulated emission depletion (STED) microscopy provides diffraction-unlimited imaging resolution while preserving the merits of confocal fluorescence microscopy, such as optical sectioning, molecular specificity, and fast imaging without the need for mathematical data postprocessing[1–3,8–10]. However, further dissemination of STED microscopy has been constrained by some intrinsic issues. To compete with the ultra-fast spontaneous fluorescence, a high-power ultra-fast laser with probe-defined depletion wavelength is required, which generally leads to serious photobleaching, phototoxicity effects, and re-excitation background in the samples[11,12]. In addition, high-precision overlapping of multiple pairs of lasers is typically required for multichromatic probes and for multicolor super-resolution imaging to match their different energy states, which results in high complexity, high cost, and low usability (Fig. 1a)[13,14]. Meanwhile, the spectral overlap among the utilized dyes (typically with broad emission spectra) would also lead to cross-talk between channels[15], and thereby compromises its utility and impact.

Lanthanide-doped upconversion nanoparticles have become an important group of luminophores for diverse applications thanks to their unique excitation and emission properties, and have been employed for STED nanoscopy featuring low-power, near-infrared (NIR) laser excitation and ultrahigh photostability[16–21]. However, photo-switching in lanthanide nanoparticles is unfortunately rather limited and difficult to tune, and the depletion laser needs to directly act on the emitting activators for emissive-state depopulation, which is essentially the same as in traditional STED implemented with organic dyes and quantum dots[17,18]. Since upconversion nanoparticles comprise two types of optical centers, forming a typical binary sensitizer-activator system, suppression of the energy flow from the sensitizer to the activator would be a viable method to inhibit the emission of these multichromatic fluorophores. Here, we establish a powerful emission inhibition mechanism of cutting off the excitation chain realized in lanthanide-doped nanoparticles, named stimulated-emission induced excitation depletion (STExD) manifesting cascade amplified depletion, low depletion saturation intensity, and depletion light in the second near-infrared (NIR-II) tissue optical window. Different from the traditional depletion approach in STED of directly depleting the emitting state, the proposed STExD approach occurs in the sensitizer and thus allows emission inhibition to be achieved for many different emitters using a single depletion wavelength (Fig. 1b).

## Results

**Cascade amplified depletion reduces the saturation intensity.** To implement this, we employ $Nd^{3+}$ as the sensitizer[22] as it has a typical quasi-four-level configuration suitable for lasing[23]. As a long-lived energy level, the $^4F_{3/2}$ state of $Nd^{3+}$ is in stark contrast with the ultra-fast decaying $^4I_{11/2}$ state. Together with a large emission cross-section, a population inversion can be readily established between the $^4F_{3/2}$ and $^4I_{11/2}$ states, and the depletion saturation intensity (defined as the applied depletion intensity at which depletion efficiency (DE) reaches 50%) can be one or two orders of magnitude lower than those of organic dyes (Fig. 2a, Supplementary Fig. 1a, and Supplementary Table 1). Our numerical simulations on the effect of population inversion show that the depletion saturation intensity decreases with an increasing inverted population in this quasi-four-level system (Methods and Supplementary Fig. 1).

Guided by the results from numerical simulations, a lab-made microscopy system coupled with a NIR continuous-wave (CW) excitation beam at 740 nm[24] and a NIR-II CW depletion beam at 1064 nm was built to perform spectroscopic and imaging studies (Supplementary Fig. 2). Next, we synthesized singly-doped $NaYF_4$:Nd (3%) nanoparticles (Supplementary Figs. 3a, 4a, c and Supplementary Table 2) and experimentally investigated the optical emission inhibition characteristics of $Nd^{3+}$ ions. Upon excitation with a 740-nm laser (Supplementary Fig. 5a), ground-state absorption (GSA, $^4I_{9/2} \rightarrow ^4F_{7/2}$) was initiated for $Nd^{3+}$ ions. This is followed by three-step sequential processes of energy transfer upconversion (ETU) to the other $Nd^{3+}$ ions to generate luminescence from higher levels, including $^4F_{3/2} + ^4I_{9/2} \rightarrow ^4I_{9/2} + ^4F_{3/2}$ (ETU1), $^4F_{3/2} + ^4F_{3/2} \rightarrow ^4I_{9/2} + ^2P_{1/2}$ (ETU2), and $^4F_{3/2} + ^4G_{5/2} \rightarrow ^4I_{9/2} + ^4D_{3/2}$ (ETU3). The intensities of one-photon emissions at 867 and 900 nm ($^4F_{3/2} \rightarrow ^4I_{9/2}$), two-photon emissions at 523 nm ($^4G_{9/2} \rightarrow ^4I_{9/2}$), 533 nm ($^4G_{7/2} \rightarrow ^4I_{9/2}$), 573 nm ($^4G_{5/2} \rightarrow ^4I_{9/2}$), 588 nm ($^2P_{1/2} \rightarrow ^4I_{15/2}$), 640 nm ($^4G_{11/2} \rightarrow ^4I_{15/2}$), and 660 nm ($^2G_{9/2} \rightarrow ^4I_{15/2}$), and three-photon emissions at 415 nm ($^2P_{3/2} \rightarrow ^4I_{11/2}$) and 450 nm ($^4D_{3/2} \rightarrow ^4I_{15/2}$) were simultaneously collected (Fig. 2b). Their nonlinearity was confirmed by the measured power dependence of the emission intensity (Supplementary Fig. 6). When the 1064-nm laser was applied, all emission bands were sharply inhibited, which can be attributed to the stimulated emission process ($^4F_{3/2} \rightarrow ^4I_{11/2}$) induced by the 1064-nm laser as discussed above. Importantly, the re-excitation background by the depletion beam was not observed here (Fig. 2b and Supplementary Fig. 5a, b). It is worth noting that the depletion efficiencies were found to be different for different emission bands. With the co-irradiation of the 1064-nm beam (2.5 MW cm$^{-2}$), relatively low depletion efficiency for the

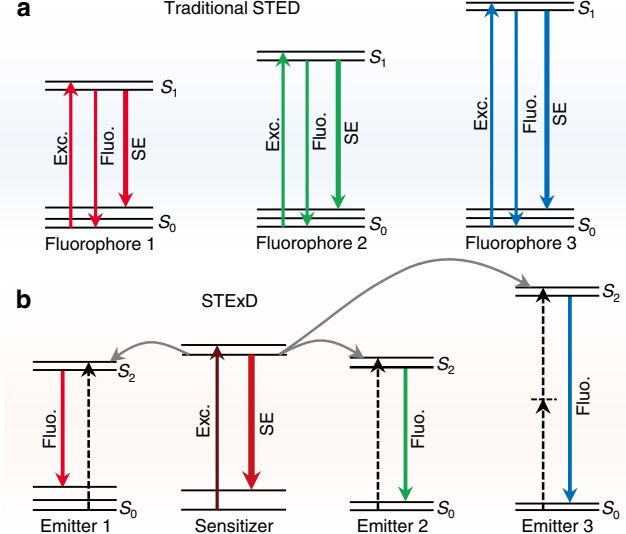

**Fig. 1 Schematics of two distinct emission inhibition mechanisms. a** In traditional STED, the wavelengths of the excitation and depletion laser beams typically have to be carefully adapted with the varied spectroscopic properties of multi-chromatic probes. Multiple pairs of laser beams have to be temporally and spatially overlapped with high-precision for multicolor imaging. For multi-chromatic imaging, a unified excitation or depletion light wavelength would lead to low excitation or depletion efficiency and possible channel cross-talk. **b** The proposed stimulated-emission induced excitation depletion (STExD) strategy can generate depopulation simultaneously for multiple emitting states of multi-chromatic probes by utilizing a single depletion wavelength to de-excite their common sensitizer. In STExD, the depletion laser is defined by the common sensitizer, and thereby the strong dependence of the depletion wavelength on the emitting states and emission colors will be abnormally broken.

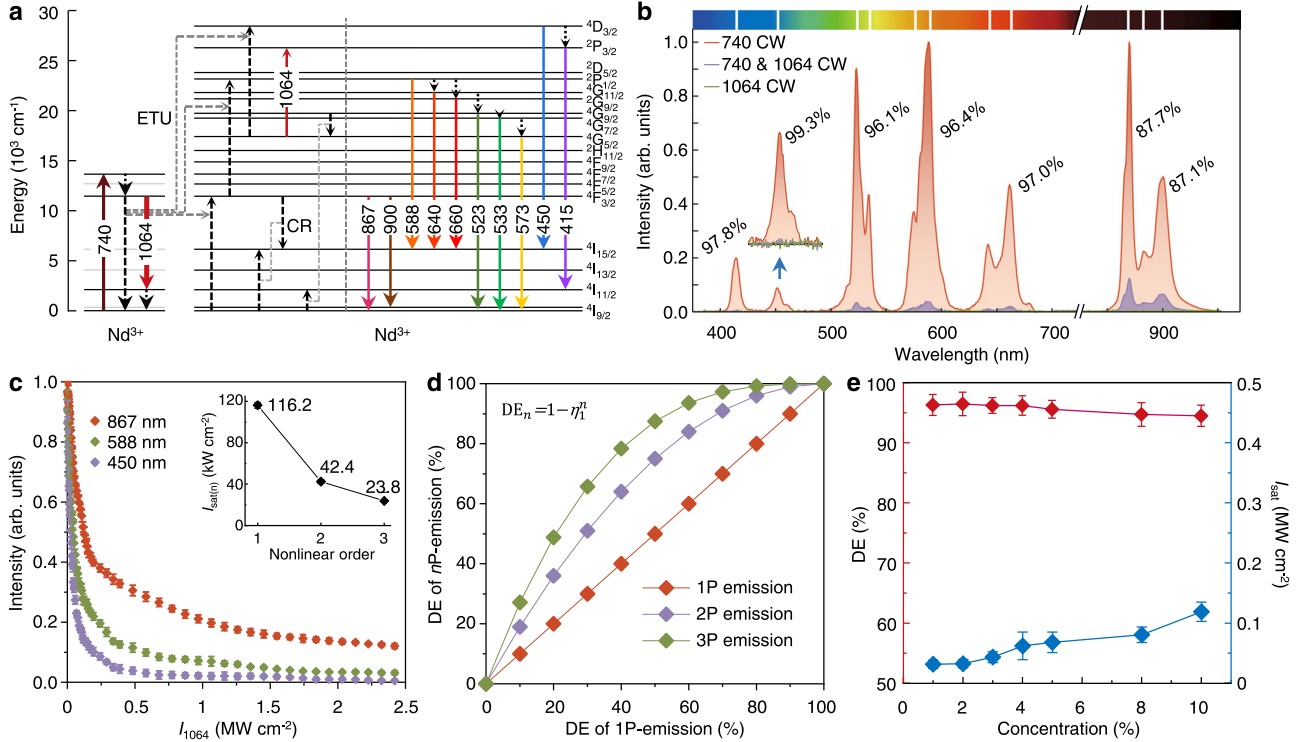

**Fig. 2 Cascade amplified depletion of STExD in NaYF$_4$:Nd nanoparticles. a** Energy level diagram of Nd-upconversion system illustrating the emission inhibition process with the 1064-nm beam. **b** Full-spectrum emission inhibition, and the depletion efficiencies of different emissions of NaYF$_4$:Nd (3%) nanoparticles under 740 and 1064 nm co-irradiation. **c** Emission intensity of one-photon (867 nm), two-photon (588 nm), and three-photon (450 nm) emissions from NaYF$_4$:Nd (3%) nanoparticles versus depletion laser intensity ($I_{740} = 78.3$ kW cm$^{-2}$, $I_{1064} = 0$–2.5 MW cm$^{-2}$). Inset: the measured values of $I_{sat}$ were 116.2 ± 1.3, 42.4 ± 0.7, and 23.8 ± 0.4 kW cm$^{-2}$ of 867, 588, and 450 nm emissions, respectively. **d** Comparison between calculated depletion efficiencies (DE, emission intensity ratio ($I_{740}$-$I_{740\&1064}$)/$I_{740}$) of one-, two-, and three-photon emissions. **e** The measured depletion efficiency and saturation intensity ($I_{sat}$, 50% emission off) of 588 nm emission versus the Nd$^{3+}$ concentration. NaYF$_4$:Nd (x%) nanoparticles with x = 1, 2, 3, 4, 5, 8, 10. Data were presented as mean values ± standard deviation (SD). Error bars are defined as the SD of $n = 3$ independent measurements.

one-photon emission at 867 and 900 nm was observed, approximately 87%. In contrast, two-photon emission at 588 nm and three-photon emission at 450 nm exhibited much higher depletion efficiencies of up to 97.8 ± 0.5% and 99.3 ± 0.3%, respectively (Fig. 2b and Supplementary Table 3). The slight discrepancy in depletion efficiency between 415 and 450 nm emissions may be attributed to the excited state absorption (ESA) process of $^4G_{5/2} \rightarrow {}^2P_{3/2}$ driven by the depletion beam. The depletion saturation intensity of the three-photon emission at 450 nm was determined to be 23.8 ± 0.4 kW cm$^{-2}$, nearly five times lower than that of the one-photon emission at 867 nm, and three orders of magnitude lower than that of traditional STED fluorophores[25,26] (Fig. 2c).

We ascribed this difference to a cascade amplified depletion effect, originating from the stepwise ETU process. As the stimulated emission process directly depleted the $^4F_{3/2}$ state, the depletion efficiency of one-photon emissions would be equivalent to the percentage of the electrons depleted from the $^4F_{3/2}$ state. However, since pumping an ion to the two-photon emitting state i.e., $^2P_{1/2}$, needs to consume two electrons at the excited state $^4F_{3/2}$, the population in $^2P_{1/2}$ would exhibit a squared relationship to that of the $^4F_{3/2}$ state, which leads to a much higher depletion efficiency for the two-photon emission than for the one-photon emission, and similarly also for the three-photon emission. Thus, higher-order emissions relying on multiple energy transfer processes will be depleted more efficiently with the effect of cascade amplified depletion. For example, assuming that one-photon emission has a depletion efficiency of 50%, the depletion efficiencies for two-photon and three-photon emissions rapidly increase to 75 and 87.5%, respectively (Supplementary Fig. 7a).

Theoretically, the depletion efficiency of an $n$-photon emission process can be calculated as DE$_n = 1 - \eta_1^n$, where $\eta$ denotes the remaining ratio of one-photon emission intensity in the presence of a depletion beam, leading to a higher depletion rate ($\eta^{-1}$) of multiphoton emissions (Methods, Fig. 2d and Supplementary Fig. 7). The alleviation in saturation intensity enabled by the cascade amplified depletion was consistent with our theoretical expectations and verified by numerical simulations (Methods and Supplementary Figs. 8, 9). When increasing the doping concentration (1–10%), the high depletion efficiency was hardly affected and the saturation intensities were slightly elevated (Fig. 2e), which could be ascribed to the fact that the enhanced cross-relaxation (CR) process of $^4G_{9/2} + {}^4I_{9/2} \rightarrow {}^4G_{5/2} + {}^4I_{11/2}$ would compete with the stimulated emission process (Fig. 2e and Supplementary Fig. 10a, b). In addition, the brightness of particles first increases and then saturates with increasing doping concentration, governed by the laser absorption and the CR process of $^4F_{3/2} + {}^4I_{9/2} \rightarrow {}^4I_{15/2} + {}^4I_{15/2}$[27,28] (Supplementary Fig. 10c, d). To investigate the optical depletion kinetics, the time-resolved intensity curves were acquired through modulating excitation beam or depletion beam. With the co-irradiation of the pulsed 740-nm beam and CW 1064-nm beam, the measured decay lifetime of the $^2P_{1/2}$ state decreased from 30.8 ± 0.5 to 12.7 ± 0.2 µs when increasing the depletion intensity (Supplementary Fig. 11a), suggesting the effect of the sensitizing $^4F_{3/2}$ states, greatly quenched by stimulated emission, on the decay time of this higher-lying level[18,29,30]. At the exaction of the CW 740-nm beam, the emission can be obviously inhibited by the pulsed 1064-nm beam (Supplementary Fig. 11b). The time

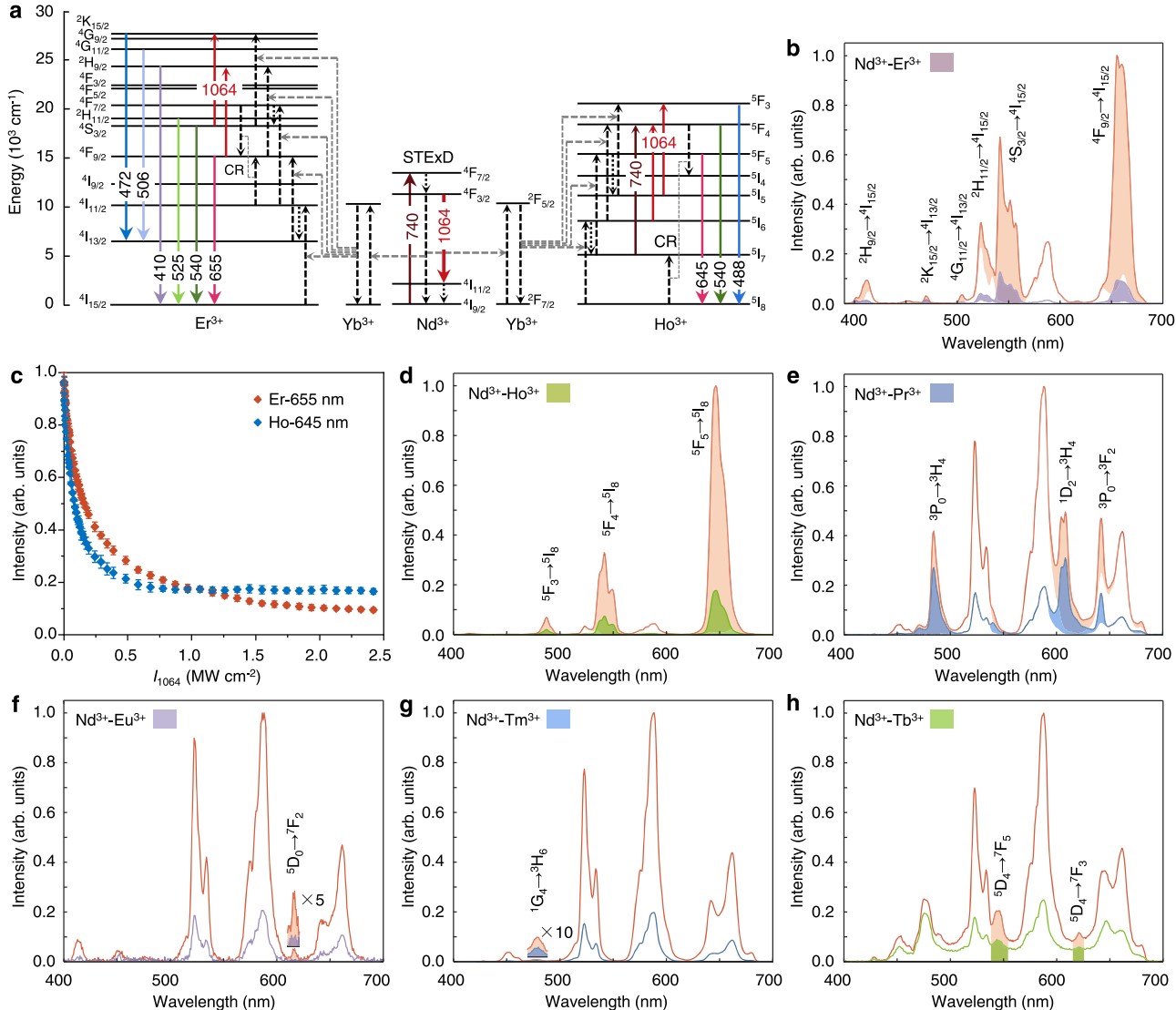

**Fig. 3 STExD strategy extended to various Nd-sensitized nanoparticles. a** Optical emission inhibition mechanisms in Nd/Yb/Er and Nd/Yb/Ho systems. **b** Full-spectrum emission inhibition of NaYF$_4$:Nd/Yb/Er (3/1/5%) nanoparticles under 740 and 1064 nm co-irradiation ($I_{740}$ = 78.3 kW cm$^{-2}$; $I_{1064}$ = 2.5 MW cm$^{-2}$). **c** Emission intensity at the 655 nm of Er$^{3+}$ and the 645 nm of Ho$^{3+}$ versus the depletion intensity ($I_{740}$ = 78.3 kW cm$^{-2}$; $I_{1064}$ = 0–2.5 MW cm$^{-2}$). The measured $I_{sat}$ were 160.9 ± 1.7 (Er-655 nm) and 85.8 ± 0.7 (Ho-645 nm) kW cm$^{-2}$. Data were presented as mean values ± SD. Error bars are defined as the SD of $n$ = 3 independent measurements. **d–h** Full-spectrum emission inhibition of other nanoparticles NaYF$_4$:Nd/Yb/Ho (3/0.5/2%), Nd/Pr (3/0.5%), Nd/Eu (3/15%), Nd/Yb/Tm (3/1/0.2%), and Gd/Tb@Gd/Nd/Yb/Tm (50/15@50/3/5/1%) under 740 and 1064 nm co-irradiation.

required to reach a new steady-state decreased from 51.7 ± 1.9 to 13.8 ± 0.1 μs with the increasing depletion intensities.

**A single NIR-II CW laser enables STExD for multi-chromatic emitters**. The verified emission inhibition in the Nd$^{3+}$ self-sensitized system lays the groundwork for extending the STExD strategy to other Nd$^{3+}$-sensitized luminescence systems[22]. As proof-of-concept experiments, a series of Nd$^{3+}$-sensitized nanoparticles were prepared and studied. We first characterized the emission inhibition performance of NaYF$_4$: Nd/Yb/Er (3/1/5%) nanoparticles (Supplementary Figs. 3b, 4b, c). As shown in Fig. 3a, upon excitation at 740 nm, the activator Er$^{3+}$ obtained excitation energy from Yb$^{3+}$ and Nd$^{3+}$ via multistep energy transfer, displaying characteristic emissions centered at 655 nm ($^4F_{9/2} \rightarrow {}^4I_{15/2}$), 540 nm ($^4S_{3/2} \rightarrow {}^4I_{15/2}$), 525 nm ($^2H_{11/2} \rightarrow {}^4I_{15/2}$), 506 nm ($^4G_{11/2} \rightarrow {}^4I_{13/2}$), 472 nm ($^2K_{15/2} \rightarrow {}^4I_{13/2}$), and 410 nm

($^2H_{9/2} \rightarrow {}^4I_{15/2}$)[31] (Fig. 3b and Supplementary Fig. 12a), and the 588 nm emission of Nd$^{3+}$ is also present[32]. When the 1064-nm depletion beam was applied, the de-excitation effect imposed on Nd$^{3+}$ sensitizers indirectly depleted the emitting states of the activators. We unmixed the emission spectrum of Er$^{3+}$ from Nd$^{3+}$ by normalizing the 588 nm emission band. When Er$^{3+}$-activated nanoparticles were excited by the 740-nm laser (non-resonant with Er$^{3+}$) instead of the 800-nm laser (resonant with Er$^{3+}$, Supplementary Fig. 5c), the depletion efficiency of 655 nm emission was measured to be 90.3 ± 0.4% (Fig. 3b, c and Supplementary Table 3), which by far exceeds the previously reported depletion efficiency of 27% in Er$^{3+}$-activated nanoparticles[16], with a saturation intensity of 160.9 ± 1.7 kW cm$^{-2}$. The higher depletion efficiency of the 655 nm emission than that of the emissions from higher states was attributed to the 1064-nm laser-induced ESA processes of $^4F_{9/2} \rightarrow {}^2H_{9/2}$ and $^4S_{3/2} \rightarrow {}^2K_{15/2}$. The STExD effect in

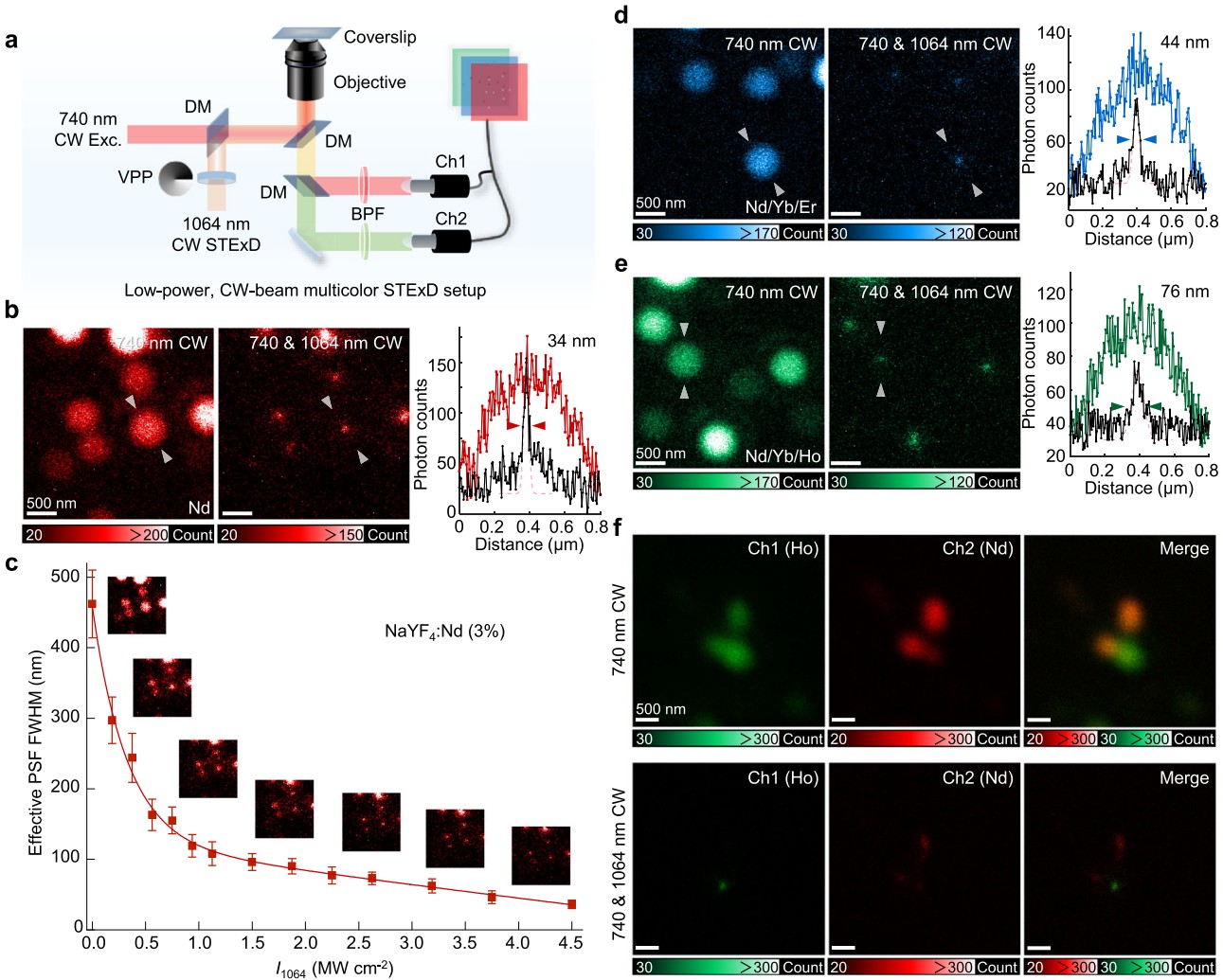

**Fig. 4 STExD imaging for multi-chromatic probes and two-color STExD. a** Simplified schematic illustrating the STExD super-resolution microscopy system with two-channel detection. **b** Super-resolution imaging and line profile analyses of NaYF$_4$:Nd (3%) nanoparticles (16.7 ± 0.9 nm). **c** Measured effective PSF FWHM plotted versus the depletion intensity ($I_{740}$ = 78.3 kW cm$^{-2}$; $I_{1064}$ = 0–4.5 MW cm$^{-2}$). Data were presented as mean values ± SD. Error bars are defined as the SD of $n$ = 3 independent measurements. **d** Super-resolution imaging and line profile analyses of NaYF$_4$:Nd/Yb/Er (3/1/5%) nanoparticles (20.5 ± 1.0 nm). **e** Super-resolution imaging and line profile analyses of NaYF$_4$: Nd/Yb/Ho (3/0.5/2%) nanoparticles (21.3 ± 0.9 nm). **f** Two-color super-resolution imaging of NaYF$_4$:Nd/Ce (3/20%, 17.9 ± 1.2 nm) and Nd/Yb/Ho (3/0.5/2%, 21.3 ± 0.9 nm) nanoparticles. The 645 nm emission from NaYF$_4$:Nd/Yb/Ho was captured by channel 1 (in green). The 588 nm emissions from NaYF$_4$:Nd/Ce were captured by channel 2 (in red). $I_{740}$ = 78.3 kW cm$^{-2}$; $I_{1064}$ = 4.5 MW cm$^{-2}$.

Nd/Yb/Er system was also supported by theoretical calculations (Supplementary Note 1 and Supplementary Figs. 13a, 14a).

We next verified STExD in NaYF$_4$:Nd/Yb/Ho (3/0.5/2%) nanoparticles (Supplementary Figs. 3c, 4c). Co-irradiation with a 1064-nm laser beam efficiently quenched the 645 nm ($^5F_5 \rightarrow {}^5I_8$), the 540 nm ($^5F_4 \rightarrow {}^5I_8$), and the 488 nm ($^5F_3 \rightarrow {}^5I_8$) emission bands of Ho$^{3+}$ originally generated by the 740-nm laser (Fig. 3a, d, Supplementary Fig. 5d, and Supplementary Table 3). To the best of our knowledge, this is the first report of optically controlled emission inhibition in Ho$^{3+}$-activated nanoparticles. These results were also in accordance with theoretical calculations (Supplementary Note 2 and Supplementary Figs. 13b, 14b). The depletion efficiency of the 645 nm emission was ~82.2 ± 0.3%, with a saturation intensity of 85.8 ± 0.7 kW cm$^{-2}$ (Fig. 3c). The ESA processes of $^5I_7 \rightarrow {}^5F_4$, $^5I_6 \rightarrow {}^5F_4$ (phonon-assisted), and $^5I_5 \rightarrow {}^5F_3$ compromised the depletion efficiencies of Ho$^{3+}$ emissions, i.e., 650, 540, and 488 nm emissions, respectively. As an extensible mechanism, the STExD was also found to be applicable to other Nd-sensitized systems, including NaYF$_4$:Nd/Pr,

Nd/Eu, Nd/Yb/Tm, and NaYF$_4$:Gd/Tb@ Gd/Nd/Yb/Tm nano-particles (Supplementary Fig. 15). The effect of emission inhibition was achieved for Pr$^{3+}$ (642, 610, and 485 nm emissions), Eu$^{3+}$ (615 nm emission), Tm$^{3+}$ (475 nm emission), and Tb$^{3+}$ (620 and 545 nm emissions) (Fig. 3e–h and Supplementary Table 3). The observed depletion efficiencies of the emissions from Pr$^{3+}$ and Tm$^{3+}$ were relatively low due to the ESA processes of 740 nm ($^3F_2 \rightarrow {}^3P_0$ of Pr$^{3+}$) and 1064 nm ($^3F_2 \rightarrow {}^1D_2$ of Pr$^{3+}$ and $^3F_4 \rightarrow {}^3F_2$ of Tm$^{3+}$) (Supplementary Fig. 15).

**Two-color STExD imaging implemented with two low-power NIR CW lasers**. The versatile STExD mechanism described here allows applications in super-resolution fluorescence microscopy. A STExD microscope equipped with a 740-nm Gaussian excitation beam and a 1064-nm donut-shaped depletion beam was built to perform super-resolution imaging using different Nd$^{3+}$-sensitized probes (Fig. 4a and Supplementary Fig. 2). Using such a pair of fixed NIR wavelength, low-power CW lasers, we achieved super-resolution imaging of multi-chromatic fluorophores by

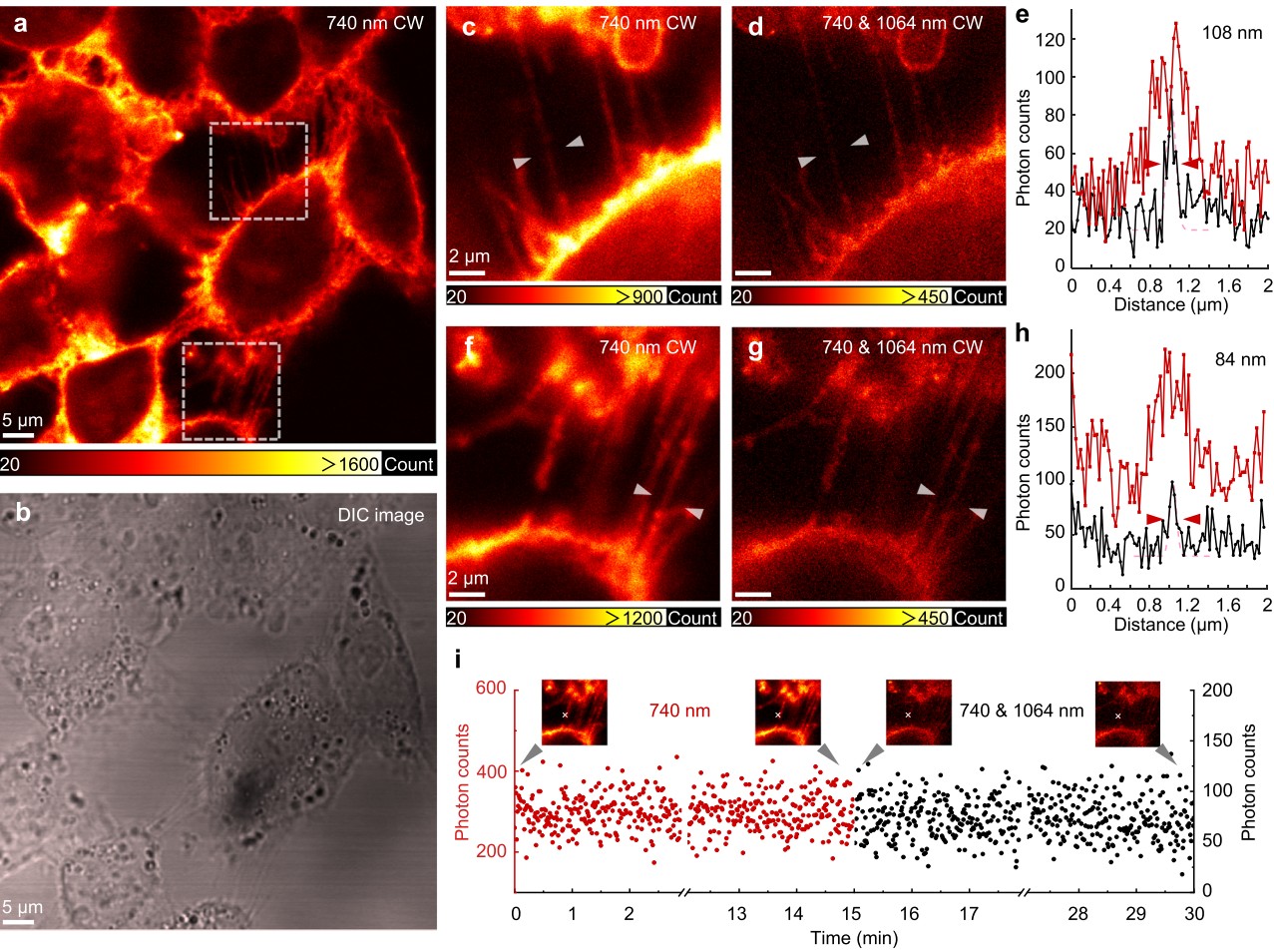

**Fig. 5 STExD imaging of immunolabelled subcellular filaments. a** Fluorescence image of actin protein in HeLa cells immunolabeled with phalloidin conjugated NaYF$_4$:Nd (3%, 10.3 ± 1.0 nm) nanoparticles under 740 nm Gaussian beam excitation. Pixel dwell time: 100 μs. $I_{740}$ = 78.3 kW cm$^{-2}$. **b** Differential Interference Contrast (DIC) image of the same sample. Three independent samples were used for this experiment with similar results obtained. **c, f** Magnified areas from white-dotted squares in (**a**). **d, g** Super-resolution imaging of the same regions as in (**c, f**) under co-irradiation with the 740 nm Gaussian beam and the 1064 nm donut-shaped beam. $I_{740}$ = 78.3 kW cm$^{-2}$; $I_{1064}$ = 3.0 MW cm$^{-2}$. **e, h** Analyses of line profiles, indicated with white arrows in (**c, d**), as well as in (**e, f**), respectively. **i** Photon counts were recorded with and without the co-irradiation of the 1064 nm donut-shaped beam. Insets show the corresponding images with the measured positions marked with white crosses.

simply switching detection filters. As a demonstration, NaYF$_4$:Nd nanoparticles were firstly imaged with visible emission bands from 500 to 700 nm. The sub-diffraction PSF FWHM of 34 nm was achieved, and a sequence of super-resolution images with low depletion intensities were captured (Fig. 4b, c). In addition, the red emission bands of NaYF$_4$:Nd/Yb/Er and NaYF$_4$:Nd/Yb/Ho nanoparticles were also imaged with obtained effective PSFs FWHM of 44 nm and 76 nm, respectively (Fig. 4d, e). In addition to the ability to image different probes with a single pair of simple, fixed-wavelength, and low-power NIR lasers, another capability is to perform imaging of multiple targets by multicolor super-resolution imaging. To demonstrate this, we performed two-color STExD imaging using these Nd$^{3+}$-sensitized nanoparticles. The 645 nm red emission from the Ho$^{3+}$ emitters was chosen as the channel 1 signal and the 588 nm emission from Nd$^{3+}$ emitters as channel 2. To this end, Ce$^{3+}$ ions were added as co-dopants to quench the 645 nm emission of Nd$^{3+}$ (Supplementary Fig. 16). Due to the narrow emission band feature of lanthanide ions, the emission bands of interest can be easily separated with a dichroic mirror. The top row of Fig. 4f displays the diffraction-limited microscopic images of two channels and their superimposed image. It is evident that the two types of

nanoparticles can be clearly distinguished in different channels, when subject to one and the same 740-nm excitation beam. Under co-irradiation with a 1064-nm beam, the effective PSFs in both channels were significantly narrowed.

**STExD imaging of immunolabelled subcellular filaments.** These nanoprobes naturally allow super-resolution imaging based on cellular immunolabeling. As a proof of concept, we imaged subcellular actin protein structures of HeLa cells, where NaYF$_4$:Nd (3%) nanoparticles were surface modified with polyacrylic acid (PAA) to acquire hydrophilicity and biocompatibility, and then conjugated with amino-phalloidin molecules (Supplementary Fig. 17). Fixed HeLa cells were incubated with these phalloidin modified nanoparticles and imaged under the STExD microscope. The actin fibers of the HeLa cells were clearly observed under the excitation at 740 nm (Fig. 5a, b). The spatial resolution was significantly improved with co-irradiation at 1064 nm. Two regions of interest from the imaged sample were magnified, and line profiles were fitted with Gaussian functions (Fig. 5c–h). The FWHMs of the line profiles were below 100 nm, revealing fine structures of the subcellular actin filaments.

Furthermore, the brightness of these nanoprobes did not show any decrease over 30-min laser scanning imaging, exhibiting excellent photostability (Fig. 5i).

## Discussion

In this study, we proposed and demonstrated the STExD strategy with cascade amplified depletion for effective emission inhibition of a variety of lanthanide emitters and for multicolor super-resolution imaging with one fixed pair of CW NIR lasers. The concept of STExD properly combines the physical processes of sensitization and stimulated emission, and makes full use of two upconversion characteristics: the binary sensitizer-activator system and the cascade multiphoton process. The continuous development of new design and synthesis methods is expected to produce even smaller and brighter nanoparticles, which will further enhance the detection signal-to-noise ratio, imaging resolution, and immunolabeling effect for diverse applications. The imaging speed can also be further increased by employing multi-focus or spinning disk scanning modes[33].

This STExD strategy will provide a rational method to overcome some inherent limitations associated with traditional STED techniques, such as high-intensity and photobleaching, inflexibility of lasers scheme, and complicated optical design for multicolor imaging. The employed low-intensity-level CW lasers of NIR excitation and NIR-II depletion, combining the merits of the multiphoton excitation process, hold a potential for low-photo-toxicity, deep-tissue super-resolution imaging in vivo[34,35]. This STExD mechanism can also circumvent the constraints of very limited photo-switching upconversion activators for the emerging upconversion STED super-resolution techniques[17,18]. One can also extend the STExD strategy to down-conversion lanthanide-doped nanoparticles and other sensitizer-activator (donor-acceptor) nanosystems[36,37] (e.g., Föster resonance energy transfer and organic-inorganic hybrid luminescence systems), expanding the bandwidth for multichannel super-resolution imaging. We expect that our study will lend new and exciting fundamental insights into multiplexed STED mechanism with low-power NIR-II depletion, expanding broad applications in diffraction-unlimited optical sensing, optogenetics manipulation[19], optical data storage, and lithography[38,39].

## Methods

**Synthesis of NaYF$_4$:Nd (3%) nanoparticles**. NaYF$_4$:Nd (3%) nanoparticles ($10.6 \pm 1.0$ nm) were synthesized using previously reported methods[40]. In a typical procedure, 5 mL Ln(CH$_3$CO$_2$)$_3$ aqueous solution with 0.03 mmol Nd$^{3+}$ and 0.97 mmol Y$^{3+}$ was added into a 100-mL round bottom flask containing 7.5 mL OA and 17.5 mL ODE. The mixture under stirring was heated to 120 °C for 10 min to remove water, then further heated to 150 °C for 40 min to form the lanthanide-oleate precursor. After cooling down to 40 °C, 0.148 g NH$_4$F and 0.761 g sodium oleate were added to the flask. Subsequently, the mixture was stirred for at least 2 h at 40 °C, then heated to 100 °C under a vacuum for 30 min, and then filled with argon. Under anhydrous and oxygen-free conditions, the mixture was heated to 270 °C and incubated for 1.5 h, and then cooled down to room temperature. The nanoparticles were precipitated by the addition of 15 mL ethanol and collected by centrifugation at 7500 r.p.m (relative centrifuge force (RCF) = 4788×$g$) for 5 min. The obtained nanoparticles were washed several times using ethanol and cyclohexane and were finally re-dispersed into 8 mL cyclohexane for subsequent use. By varying the concentration of Nd$^{3+}$ and Y$^{3+}$, a series of NaYF$_4$:Nd (x%) nanoparticles with x = 1, 2, 4, 5, 8, and 10 were obtained by the same procedure.

NaYF$_4$:Nd (3%) nanoparticles ($16.7 \pm 0.9$ nm) were synthesized using a similar method. After cooling the solution down to 40 °C, 10 mL of NH$_4$F-methanol solution (0.4 M) and 2.5 mL of NaOH-methanol solution (1.0 M) were mixed and quickly injected into the flask. The same following steps as in the synthesis of NaYF$_4$:Nd (3%) nanoparticles ($10.6 \pm 1.0$ nm) were repeated. NaYF$_4$:Nd/Ce (3/20%) nanoparticles ($17.9 \pm 1.2$ nm) were synthesized using similar methods as NaYF$_4$:Nd (3%) nanoparticles ($16.7 \pm 0.9$ nm).

**Synthesis of NaYF$_4$:Nd/Yb/Er (3/1/5%) nanoparticles**. A similar method was used to synthesize the NaYF$_4$:Nd/Yb/Er (3/1/5%) nanoparticles ($8.4 \pm 1.0$ nm). In a typical procedure, 5 mL Ln(CH$_3$CO$_2$)$_3$ aqueous solution with 0.03 mmol Nd$^{3+}$,

0.01 mmol Yb$^{3+}$, 0.05 mmol Er$^{3+}$, and 0.91 mmol Y$^{3+}$ was added, and the same following steps as in the synthesis of NaYF$_4$:Nd (3%) nanoparticles ($10.6 \pm 1.0$ nm) were repeated. NaYF$_4$:Nd/Yb/Er (3/1/5%) nanoparticles ($20.5 \pm 1.0$ nm) were synthesized using similar methods as NaYF$_4$:Nd (3%) nanoparticles ($16.7 \pm 0.9$ nm) in addition to changing the final reaction temperature to 300 °C.

**Synthesis of NaYF$_4$:Nd/Yb/Ho (3/0.5/2%) nanoparticles**. For the synthesis of NaYF$_4$:Nd/Yb/Ho (3/0.5/2%) nanoparticles ($9.1 \pm 1.0$ nm), 5 mL Ln(CH$_3$CO$_2$)$_3$ aqueous solution with 0.03 mmol Nd$^{3+}$, 0.005 mmol Yb$^{3+}$, 0.02 mmol Ho$^{3+}$, and 0.945 mmol Y$^{3+}$ was added, and the same following steps as in the synthesis of NaYF$_4$:Nd (3%) nanoparticles ($10.6 \pm 1.0$ nm) were repeated. NaYF$_4$:Nd/Yb/Ho (3/0.5/2%) nanoparticles ($21.3 \pm 0.9$ nm) were synthesized using similar methods as NaYF$_4$:Nd/Yb/Er (3/1/5%) nanoparticles ($20.5 \pm 1.0$ nm).

**Synthesis of NaYF$_4$:Nd/Pr (3/0.5%) nanoparticles**. For the synthesis of NaYF$_4$:Nd/Pr (3/0.5%) nanoparticles ($9.6 \pm 0.9$ nm), 5 mL Ln(CH$_3$CO$_2$)$_3$ aqueous solution with 0.03 mmol Nd$^{3+}$, 0.005 mmol Pr$^{3+}$, and 0.965 mmol Y$^{3+}$ was added, and the same following steps as in the synthesis of NaYF$_4$:Nd (3%) nanoparticles ($10.6 \pm 1.0$ nm) were repeated.

**Synthesis of NaYF$_4$:Nd/Eu (3/10%) nanoparticles**. For the synthesis of NaYF$_4$:Nd/Eu (3/10%) nanoparticles ($10.4 \pm 1.1$ nm), 5 mL Ln(CH$_3$CO$_2$)$_3$ aqueous solution with 0.03 mmol Nd$^{3+}$, 0.1 mmol Eu$^{3+}$, and 0.87 mmol Y$^{3+}$ was added, and the same following steps as in the synthesis of NaYF$_4$:Nd (3%) nanoparticles ($10.6 \pm 1.0$ nm) were repeated.

**Synthesis of NaYF$_4$:Nd/Yb/Tm (3/1/0.2%) nanoparticles**. For the synthesis of NaYF$_4$:Nd/Yb/Tm (3/1/0.2%) nanoparticles ($8.9 \pm 0.9$ nm), 5 mL Ln(CH$_3$CO$_2$)$_3$ aqueous solution with 0.03 mmol Nd$^{3+}$, 0.01 mmol Yb$^{3+}$, 0.002 mmol Tm$^{3+}$, and 0.958 mmol Y$^{3+}$ was added, and the same following steps as in the synthesis of NaYF$_4$:Nd (3%) nanoparticles ($10.6 \pm 1.0$ nm) were repeated.

**Synthesis of NaYF$_4$:Gd/Tb (50/15%)@NaYF$_4$:Gd/Nd/Yb/Tm (50/3/5/1%) nanoparticles**. The NaYF$_4$:Gd/Tb (50/15%)@NaYF$_4$:Gd/Nd/Yb/Tm (50/3/5/1%) core-shell nanoparticles ($18.4 \pm 1.4$ nm) were synthesized using a previously reported protocol with some modification[41]. Firstly, NaYF$_4$:Gd/Tb (50/15%) core nanoparticles ($12.8 \pm 1.1$ nm) were synthesized following steps in the synthesis of NaYF$_4$:Nd (3%) nanoparticles ($16.7 \pm 0.9$ nm), except with changing the dopants to 0.5 mmol Gd$^{3+}$, 0.15 mmol Tb$^{3+}$, and 0.35 mmol Y$^{3+}$. Another 5 mL Ln(CH$_3$CO$_2$)$_3$ aqueous solution with 0.5 mmol Gd$^{3+}$, 0.03 mmol Nd$^{3+}$, 0.05 mmol Yb$^{3+}$, 0.01 mmol Tm$^{3+}$, and 0.41 mmol Y$^{3+}$, was added into a 100-mL round bottom flask containing 7.5 mL OA and 17.5 mL ODE. The solution was heated to 120 °C for 10 min to remove water, then further heated to 150 °C for 40 min to form the lanthanide-oleate precursor. Subsequently, the solution was cooled to 80 °C. The 8-mL suspension of bare core NaYF$_4$:Gd/Tb (50/15%) nanoparticles ($12.8 \pm 1.1$ nm) was pipetted into the flask and kept at this temperature for 30 min to remove cyclohexane. After the solution cooled down to 40 °C, the following steps were kept the same as the synthesis of core nanoparticles.

**Ligand-exchange for PAA-nanoparticles**. The ligand-exchange process of nanoparticles was prepared using previously reported methods[42]. Typically, 5 mL DMF and 50 mg NOBF$_4$ were added into 5 mL cyclohexane containing 0.1 mmol as-prepared nanoparticles and vigorously stirred for 5 min. The nanoparticles were precipitated by adding an excess of toluene and collected by centrifugation at 14,000 r.p.m. (RCF = 16,684×$g$). for 20 min. The product was re-dispersed into 1 mL water and added into PAA solution (1.5 wt%) drop by drop. The mixture was stirred over 2 h and centrifugation was performed at 15000 r.p.m. (RCF = 19,152×$g$) for 20 min. The nanoparticles were re-dispersed in water to obtain a colorless and transparent colloidal solution.

**Amino-phalloidin linking to PAA-nanoparticles**. Amino-phalloidin linking with PAA-nanoparticles was carried out according to the commonly used methods[43]. In a typical experiment, 20 µL EDC (0.2 mg µL$^{-1}$, MES) and 20 µL NHS (0.3 mg µL$^{-1}$, MES) solution were prepared and added into 1 mL PAA-nanoparticles solution (1 mg mL$^{-1}$, MES). The mixture was stirred for 30 min at room temperature, and then centrifugation at 15,000 r.p.m. (RCF = 19,152×$g$) for 20 min to acquire the NHS-nanoparticles. the precipitate was immediately re-dispersed in 1 mL HEPES solution. About 10 µL amino-phalloidin DMSO solution (1 mg mL$^{-1}$, DMSO) was added to 200 µL NHS-nanoparticles solution, and the mixture was stored at 4 °C overnight for subsequent use.

**HeLa cells culture and labeling**. HeLa cells were cultured and labeled according to commonly-used culture and immunostaining techniques[26]. The HeLa cells were cultured in 96-well confocal plates with about 15,000 cells in each unit overnight. On the second day, the cells were rinsed with PBS buffer and fixed using immunostaining fixative for 20 min at room temperature. The fixed cells were washed and permeabilized with immunostaining permeate for 20 min. Washing followed

by the addition of immunostaining blocking solution for 20 min. Then, the as-prepared phalloidin-nanoparticles were dissolved in HEPES buffer after centrifugation. And a volume of 200 µL solution (200 µg mL$^{-1}$) was added. The staining reaction of actin protein was kept at 4 °C overnight. The cells were then rinsed with HEPES buffer three times for subsequent imaging.

**Characterization**. The transmission electron microscopic (TEM) imaging and the analysis of energy dispersive spectroscopy (EDS) for the as-prepared nanoparticles were characterized by a transmission electron microscope (JEOL-1400 PLUS) and analyzed with Fiji/ImageJ. The analysis of inductively coupled plasma (ICP) emission spectrometer was characterized using a plasma emission spectrometer (SPECTRO ARCOS MV). The as-prepared nanoparticles were dissolved in nitric acid and diluted to about 10 ppm for ICP analysis.

**Optical system for spectroscopic and microscopic studies**. As shown in Supplementary Fig. 2, a lab-made optical system for spectroscopic measurement and super-resolution imaging was set up. The non-descanned detectors (NDD) do not possess the function of confocal imaging. The 740 nm CW laser beam was generated by a Ti: Sapphire Oscillator (Mira HP, Coherent) and filtered with a band-pass filter (FF01-725/40-25, Semrock). A pinhole in combination with two lenses was used as a spatial filter to optimize the beam profile. The 1064 nm CW laser beam was generated by a single-mode diode laser (MAX-SPS500D, Max Photonics) and filtered with a band-pass filter (FF01-1064/5-25, Semrock). A single-mode fiber was used to optimize the beam profile. The laser beam power was controlled by two pairs of half-wave plates and polarization beam splitters placed in the optical path. These two NIR laser beams were then spatially overlapped using a dichroic mirror (T820dcspxrxt-UF2, Chroma), and then directed into a multiphoton laser scanning microscope (FV1000MPE-S with a motorized inverted stand, IX81, Olympus). In the super-resolution imaging, the 1064-nm beam was modulated by a 1064-nm vortex phase plate (VPP-m1064, RPC Photonics) to perform a 2π helix phase, generating a donut-shaped beam, and was made circularly polarized by another quarter-wave plate attached to the objective lens. The coaxial laser beams were reflected by a short-pass dichroic mirror (T690spxxf-UF1, Chroma), and focused on the sample through an oil-immersed objective (OL1, UPLXPO100XO, ×100/1.45, Olympus). A spectrometer (QE65000, Ocean Optics) was used for spectroscopic studies, where the visible emission (400–700 nm) was collected through the same objective and filtered by a short-pass filter (FF02-694/sp-25, Semrock). Photomultiplier tubes (PMT) were employed for imaging studies. Another objective (OL2, ×20/0.5, Olympus) was set on the condenser holder above the sample to collect the NIR emission, which was filtered by an 800 nm long-pass filter, and then detected by another spectrometer (QE65000, Ocean Optics). The two spectrometers can collect the spectra signal from the same sample region of interest simultaneously. A pair of polarization beam splitters and half-wave plates were placed in the optical path to adjust the laser powers, which were recorded by a power meter by using a pellicle beam splitter to reflect 5% of laser power. The size of the laser focus spot was defined as the full width at 1/e$^2$ of maximum, and the laser intensities were calculated by dividing the powers by the area of laser spots.

**Luminescence rise/decay times measurement**. For the measurement of luminescence rise time and decay time, before coupling in the two beams, the 740 nm laser beam was modulated with a chopper (model SR540, Stanford), and the time-resolved luminescence intensity was recorded to perform the emission decay time measurements of the nanoparticles. The emission photons were selected by different filters and were detected by a PMT. The trigger signal from the chopper was synchronized with a time-correlated single-photon counter (NanoHarp, Picoquant). The chopper modulation frequency was 500 Hz. The chopper was placed in the focal plane where the beam was tightly focused to shorten the falling edge of the laser pulse.

**Multicolor super-resolution imaging**. In the two-color super-resolution microscopic imaging, a short-pass dichroic mirror (610spxxr-UF1, Chroma) was used to separate the luminescence to fall onto two PMT detectors. A band-pass filter (ET645/30x, Chroma) was placed in front of PMT1 to collect the 645 nm emission signal of NaYF$_4$:Nd/Yb/Ho (3/0.5/2%) nanoparticles. A band-pass filter (ET590/33 m, Chroma) was placed in front of PMT2 to collect the 588 nm emission signal from NaYF$_4$:Nd/Ce (3/20%) nanoparticles.

**Numerical simulation on cascade amplified depletion of STExD**. The electron transition processes in lanthanide ions can be described by a set of rate equations coupling the populations of each of the energy states. Here, we numerically simulated the cascade amplified depletion in the NaYF$_4$:Nd (3%) system. The rate-equation-based theoretical analysis and simulation were performed using MATLAB (R2019a). As shown in Supplementary Fig. 8, the rate equations of the

proposed energy transfer processes were derived as follows:

$$\mathrm{Nd^{3+}(^4I_{9/2})}: \frac{dn_0}{dt} = -\rho_1\sigma_0 n_0 + w_1 n_3 n_3 + w_2 n_3 n_5 - c_1 n_0 n_3 - c_2 n_0 n_6 \\ + \beta_1 n_1 + b_{30}\frac{n_3}{\tau_3} + \frac{n_5}{\tau_5} + \frac{n_6}{\tau_6} \tag{1}$$

$$\mathrm{Nd^{3+}(^4I_{11/2})}: \frac{dn_1}{dt} = -\rho_2\sigma_{13} n_1 + \rho_2\sigma_{31} n_3 + c_2 n_0 n_6 - \beta_1 n_1 + \beta_2 n_2 + b_{31}\frac{n_3}{\tau_3} + \frac{n_9}{\tau_9} \tag{2}$$

$$\mathrm{Nd^{3+}(^4I_{15/2})}: \frac{dn_2}{dt} = 2c_1 n_0 n_3 - \beta_2 n_2 + \beta_3 n_3 + \frac{n_7}{\tau_7} + \frac{n_8}{\tau_8} + \frac{n_{10}}{\tau_{10}} \tag{3}$$

$$\mathrm{Nd^{3+}(^4F_{3/2})}: \frac{dn_3}{dt} = \rho_2\sigma_{13} n_1 - \rho_2\sigma_{31} n_3 - 2w_1 n_3 n_3 - w_2 n_3 n_5 - c_1 n_0 n_3 - \beta_3 n_3 + \beta_4 n_4 - \frac{n_3}{\tau_3} \tag{4}$$

$$\mathrm{Nd^{3+}(^4F_{7/2})}: \frac{dn_4}{dt} = \rho_1\sigma_0 n_0 - \beta_4 n_4 + \beta_5 n_5 \tag{5}$$

$$\mathrm{Nd^{3+}(^4G_{5/2})}: \frac{dn_5}{dt} = -\rho_2\sigma_{59} n_5 - w_2 n_3 n_5 + c_2 n_0 n_6 - \beta_5 n_5 + \beta_6 n_6 - \frac{n_5}{\tau_5} \tag{6}$$

$$\mathrm{Nd^{3+}(^4G_{9/2})}: \frac{dn_6}{dt} = -c_2 n_0 n_6 - \beta_6 n_6 + \beta_7 n_7 - \frac{n_6}{\tau_6} \tag{7}$$

$$\mathrm{Nd^{3+}(^4G_{11/2})}: \frac{dn_7}{dt} = -\beta_7 n_7 + \beta_8 n_8 - \frac{n_7}{\tau_7} \tag{8}$$

$$\mathrm{Nd^{3+}(^2P_{1/2})}: \frac{dn_8}{dt} = w_1 n_3 n_3 - \beta_8 n_8 + \beta_9 n_9 - \frac{n_8}{\tau_8} \tag{9}$$

$$\mathrm{Nd^{3+}(^2P_{3/2})}: \frac{dn_9}{dt} = \rho_2\sigma_{59} n_5 - \beta_9 n_9 + \beta_{10} n_{10} - \frac{n_9}{\tau_9} \tag{10}$$

$$\mathrm{Nd^{3+}(^4D_{3/2})}: \frac{dn_{10}}{dt} = w_2 n_3 n_5 - \beta_{10} n_{10} - \frac{n_{10}}{\tau_{10}} \tag{11}$$

Here $n_i$ ($i = 0$ to 10) represents the population of Nd$^{3+}$ ions on the $^4I_{9/2}$, $^4I_{11/2}$, $^4I_{15/2}$, $^4F_{3/2}$, $^4F_{7/2}$, $^4G_{5/2}$, $^4G_{9/2}$, $^4G_{11/2}$, $^2P_{1/2}$, $^2P_{3/2}$, and $^4D_{3/2}$ states, respectively. $\tau_i$ ($i = 3, 5, 6, 7, 8, 9, 10$) represents the radiative lifetimes of the $^4F_{3/2}$, $^4G_{5/2}$, $^4G_{9/2}$, $^4G_{11/2}$, $^2P_{1/2}$, $^2P_{3/2}$, and $^4D_{3/2}$ states, respectively. $\beta_i$ ($i = 1$ to 10) represents the non-radiative decay rates of the $^4I_{11/2}$, $^4I_{15/2}$, $^4F_{3/2}$, $^4F_{7/2}$, $^4G_{5/2}$, $^4G_{9/2}$, $^4G_{11/2}$, $^2P_{1/2}$, $^2P_{3/2}$, and $^4D_{3/2}$ states, respectively. $w_1$ and $w_2$ denote the coefficients of the interionic energy transfer processes: $^4F_{3/2} + ^4F_{3/2} \rightarrow ^4I_{9/2} + ^2P_{1/2}$ and $^4F_{3/2} + ^4G_{5/2} \rightarrow ^4I_{9/2} + ^4D_{3/2}$. $c_1$ and $c_2$ denotes the coefficient of cross-relaxation processes: $^4I_{9/2} + ^4F_{3/2} \rightarrow ^4I_{15/2} + ^4I_{15/2}$ and $^4I_{9/2} + ^4G_{9/2} \rightarrow ^4I_{11/2} + ^4G_{5/2}$. $\sigma_0$ denotes the absorption cross-section for the ground-state absorption process: $^4I_{9/2} \rightarrow ^4F_{7/2}$. $\sigma_{13}$ and $\sigma_{31}$ denote the absorption cross-section for the excited state absorption process: $^4I_{11/2} \rightarrow ^4F_{3/2}$ and the stimulated emission cross-section for the stimulated emission process: $^4F_{3/2} \rightarrow ^4I_{11/2}$. $\sigma_{59}$ denotes the absorption cross-section for the excited state absorption process: $^4G_{5/2} \rightarrow ^2P_{3/2}$. $\rho_1$ and $\rho_2$ are the laser intensities of the 740 and 1064 nm beams. The values used for the main parameters are tabulated in Supplementary Table 4. To be consistent with the experimental condition, the intensity of 740 nm beams is set to 78 kW cm$^{-2}$, while the intensity of the 1064 nm beam is varied from 0 to 2.0 MW cm$^{-2}$. The change in population of the different electronic energy states with increasing depletion intensities reflects the emission inhibition behavior of each of the emission bands. As shown in Supplementary Fig. 9, the populations of $^4F_{3/2}$, $^2P_{2/1}$, $^4D_{3/2}$ states, corresponding to 867, 588, and 450 nm emissions, show a sharp downtrend with increased depletion intensities, confirming the effective emission inhibition processes arising in NaYF$_4$:Nd (3%) system. The higher-order emissions have a more efficient depletion, which is well in accordance with the proposed mechanism of cascade amplified depletion. In addition, according to the simulated population of $^4F_{3/2}$ and $^4I_{11/2}$, we can easily derive the value of the population inversion in the classic four-level Nd$^{3+}$ system. By varying the decay lifetimes of the $^4F_{3/2}$ and $^4I_{11/2}$ states (the radiative lifetime of $^4F_{3/2}$ and the non-radiative lifetime of $^4I_{11/2}$), the relationship between the population inversion and the depletion saturation intensity was analyzed, and the results are shown in Supplementary Fig. 1.

**Analytic description of the cascade amplified depletion effect**. Based on the proposed STExD mechanism, the excitation depletion phenomenon could occur in any sensitizer-activator luminescence system when the stimulated emission process is inhibiting the energy of the sensitizer. Here, to better understand the STExD mechanism and the cascade amplified depletion process, a simplified sensitizer-activator system with a four-level sensitizer and four-level activator (Supplementary Fig. 7b) is used to analyze the relationship between the depletion of the activator and

the sensitizer. This system could be expressed by the steady rate equations:

$$\frac{dn_{S0}}{dt} = -\alpha_p n_{S0} + w_1 n_{S2} n_0 + w_2 n_{S2} n_1 + w_3 n_{S2} n_2 + \beta_{S1} n_{S1} \tag{12}$$

$$\frac{dn_{S1}}{dt} = \alpha_d n_{S2} - \beta_{S1} n_{S1} \tag{13}$$

$$\frac{dn_{S2}}{dt} = -\alpha_d n_{S2} - w_1 n_{S2} n_0 - w_2 n_{S2} n_1 - w_3 n_{S2} n_2 + \beta_{S3} n_{S3} \tag{14}$$

$$\frac{dn_{S3}}{dt} = \alpha_p n_{S0} - \beta_{S3} n_{S3} \tag{15}$$

$$\frac{dn_0}{dt} = -\left(\frac{dn_1}{dt} + \frac{dn_2}{dt} + \frac{dn_3}{dt}\right) \tag{16}$$

$$\frac{dn_1}{dt} = w_1 n_{S2} n_0 - w_2 n_{S2} n_1 - \beta_1 n_1 + \beta_2 n_2 - \frac{n_1}{\tau_1} \tag{17}$$

$$\frac{dn_2}{dt} = w_2 n_{S2} n_1 - w_3 n_{S2} n_2 - \beta_2 n_2 + \beta_3 n_3 - \frac{n_2}{\tau_2} \tag{18}$$

$$\frac{dn_3}{dt} = w_3 n_{S2} n_2 - \beta_3 n_3 - \frac{n_3}{\tau_3} \tag{19}$$

In these equations, $n_{Si}$ ($i = 0$ to $3$) represents the population of sensitizer ions at different energy states. $n_i$ ($i = 0$ to $3$) represents the population of activator ions at different energy states. $\beta_i$ and $\tau_i$ ($i = 1$ to $3$) represent the non-radiative decay rates and the radiative lifetimes of different energy states in activator. $\beta_{S1}$ and $\beta_{S3}$ denote the non-radiative decay rate of the $n_{S1}$ and $n_{S3}$ state of the sensitizer. $w_i$ ($i = 1$ to $3$) denotes the energy transfer upconversion coefficients from sensitizer to activator. $\alpha_p$ and $\alpha_d$ represent the absorption rates for the excitation beam and depletion beam, respectively.

In a typical STExD system, the population of higher states would always be much smaller than that of the lower states, therefore, the non-radiative decay from higher states to lower states can be neglected. The Eqs. (17) to (19) describing the emitting states of the activator can then be rewritten as:

$$n_1 = \frac{w_1 n_{S2} n_0}{w_2 n_{S2} + \beta_1 + \frac{1}{\tau_1}} \tag{20}$$

$$n_2 = \frac{w_2 n_{S2} n_1}{w_3 n_{S2} + \beta_2 + \frac{1}{\tau_2}} \tag{21}$$

$$n_3 = \frac{w_3 n_{S2} n_2}{\beta_3 + \frac{1}{\tau_3}} \tag{22}$$

Equation (20) expresses the relation between $n_1$ and $n_{S2}$. When the value of $n_{S2}$ is decreased by the stimulated emission process in the sensitizer, two parts of the equation, $w_1 n_{S2} n_0$ and $w_2 n_{S2}$ can affect the value of $n_1$, representing the first-step and second-step ETU processes, respectively. In a typical upconversion system, the energy depletion effect caused by the second-step ETU process is always far less than the energy excitation effect caused by the first-step ETU process. Here, for clarity and simplicity, we assume that the multiphoton upconversion system is a standard multiphoton excitation system. The term $w_2 n_{S2}$ is neglected.

Therefore, we have:

$$n_1 = \frac{w_1 n_{S2} n_0}{\beta_1 + \frac{1}{\tau_1}} \tag{23}$$

with the combination of Eqs. (21) and (23), we could obtain:

$$n_2 = \frac{w_1 w_2 n_{S2}^2 n_0}{(\beta_1 + \frac{1}{\tau_1})(\beta_2 + \frac{1}{\tau_2})} \tag{24}$$

Similarly, by combining Eqs. (22) and (24), we could also get:

$$n_3 = \frac{w_1 w_2 w_3 n_{S2}^3 n_0}{(\beta_1 + \frac{1}{\tau_1})(\beta_2 + \frac{1}{\tau_2})(\beta_3 + \frac{1}{\tau_3})} \tag{25}$$

Equations (23) to (25) show an apparent relationship between each order of emission and the population of the sensitizer. The one-photon emission from $n_1$ state depends linearly on the population of $n_{S2}$ state of the sensitizer, while the two-photon and three-photon emissions from the $n_1$ and $n_2$ states appear to have a quadratic and cubic relation to the population of sensitizer. When the excited state population of the sensitizer was decreased by the stimulated emission process, it can be determined that the higher-order emissions show higher depletion than the lower-order emissions, which is in accordance with the proposed mechanism of cascade amplified depletion.

In addition, from these equations, we can approximately assume that the depletion efficiency of the one-photon emission scales with the excited state population of the sensitizer, which means that their saturation intensities are also the same. As for the two-photon emission, achieving 50% DE only requires the sensitizer to be depleted by about 30%, indicating a much lower saturation intensity for this emission than for the one-photon emission will be expected. Similarly, when the DE of the sensitizer reaches 20%, the three-photon emission already

reaches a DE of 50%. According to the formula:

$$\eta = \frac{1}{1 + \frac{I}{I_{sat}}} \tag{26}$$

where $\eta$ denotes emission depletion ratio (1-DE). $I_{sat}$ denotes depletion saturation intensity. $I$ denotes the applied depletion intensity. By determining the depletion intensity corresponding to depletion efficiencies of 20, 30, and 50% on the sensitizer emission, we found that the depletion saturation intensity of two-photon emission is effectively reduced to 40% of $I_{sat}$ of the one-photon emission, while this saturation intensity of three-photon emission is only a quarter. In conclusion, the cascade amplified depletion process in the STExD system can provide an effective way to further decrease the depletion intensity requirement. With the amplified depletion, the higher-order emissions can perform more efficient depletions than the traditional stimulated emission depletion (Fig. 2d).

**Reporting summary.** Further information on research design is available in the Nature Research Reporting Summary linked to this article.

## Data availability
The source data sets and the fluorescence spectra and super-resolution images generated and analyzed during the current study are available from the corresponding author upon request. Also available upon request are other relevant data, like schematic diagrams of the theoretical parts for this work, simulated data sets, etc.

## Code availability
The codes for theoretical modeling and numerical simulations are available from the corresponding author on request.

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

## Acknowledgements

Q.Z. acknowledges support from the National Natural Science Foundation of China (11974123, 62122028), the Guangdong Provincial Science Fund for Distinguished Yong Scholars (2018B030306015), the Guangdong Provincial Natural Science Fund Projects (2019A050510037), and the Pearl River Nova Program of Guangzhou (20171010010). H.L. acknowledges the support from a Starting Grant (2016-03804) from the Swedish Research Council (Vetenskapsrådet). J.W. acknowledges support from the Swedish Foundation for Strategic Research (SSF ITM17-0491).

## Author contributions

Q.Z. conceived and designed the project. X.G., R.P., Q.W., and Q.Z. built the optical system. Q.Z., R.P., X.G., Y.L., and H.L. were responsible for the theoretical analysis and simulation with the contributions from J.W. X.G., B.H., R.P., S.Q., L.L.-P., and U.K. were responsible for synthesis, surface functionalization, and characterization of nanoparticles. X.G., R.P., B.H., Q.W., Z.Z., S.Q., and P.Z. acquired and processed spectroscopic data. R.P. and X.G. conducted super-resolution microscopic imaging. X.G., R.P., B.H., and Q.Z. analyzed and processed the data with input from other authors. The paper was written by Q.Z., X.G., R.P., Z.Z., H.L., and J.W. with input from other authors. Q.Z. supervised the project.

## Competing interests

The authors declare no competing interests.
