## [Peer Review File · Nature Communications]

Achieving low-power single-wavelength-pair nanoscopy with NIR- II continuous-wave laser for multi-chromatic probesREVIEWER COMMENTS

Reviewer #1 (Remarks to the Author):

The diffraction limit problem has long hampered advances in microscopy towards better resolution and therefore detection of nanoscopic objects. One technique to overcome this limit is stimulated emission depletion (STED) microscopy. Since the invention of its principle in 1986 by Russian scientist Victor Okhonin, STED microscopy has however seen a slow development in view of the high-performing optical tools it needs, particularly high-power, ultrafast mode-locked lasers. Its first experimental demonstration was only made in 1999 by Stefan Hell who received the Nobel prize for it in 2014. In 2017, introduction of lanthanide-based upconversion nanoparticles (UCNPs) allowed achieving optical resolution of 28 nm at low-power excitation. Photo-switching in lanthanide nanoparticles is unfortunately rather limited and difficult to tune.

Here the authors bring a new dimension to the solution of the problem by proposing a depletion mechanism that works in the second biological window (NIR-II), a spectral range particularly useful for imaging. The idea is to transpose the depletion mechanism from the emitter to the sensitizer; in this way (STExD), subsequent emission depletion can be achieved for different emitters by using just one depletion wavelength. Here Nd(III) is used as sensitizer and after verifying emission depletion in an all-Nd(III) system, the authors successfully demonstrate its application to Nd/Yb/Ln nanoparticles (Ln = Er, Ho, Tm), Nd/Ln nanoparticles (Ln = Pr, Eu), and Gd/Tb@Gd/Nd/Yb/Tm nanoparticles. A common feature of these systems is the low saturation intensity (in the range 24-160 kW/cm²) and large depletion efficiency (the range 80-99%), overcoming by large the performances of other previously proposed systems.

As a proof-of-concept, the authors have built a nanoscope based on the STExD principle. Super-resolution images of nanoparticles have resolution down to 34 nm and immunolabelled subcellular filaments of HeLa cells could be revealed with resolution < 100 nm.

Generally speaking I find the demonstration rather convincing and the paper well and clearly written with quite didactic illustrations. The innovation presented here has great potential for diffraction-unlimited imaging both in physical and biological sciences.

Technical remarks

- I regret that elemental analysis of the nanoparticles (e.g. by EDS) is not given, particularly with respect to the doping content of the Ln ions
- Quantitative data (depletion efficiency, saturation intensity, lifetimes, a.s.o) should be associated with their experimental uncertainties.

Reviewer #2 (Remarks to the Author):

The review of an article "Achieving low-power, single-wavelength-pair nanoscopy with NIR-II CW laser for multi-chromatic probes" by Xin Guo et al.

The article presents an interesting concept of using single Nd³⁺ sensitizer ions as the starting point to Yb³⁺-Ln³⁺ (Ln=Er, Ho, Tm, Pr, Tb, Eu ³⁺ ions) upconversion. The Nd³⁺ ions, were shown to be excited under ca.800nm and undergo significant depletion under 1064 nm. The luminescence from such co-doped NPs occurs from the co-doping ions (the Ln³⁺ listed above) under the condition the proceeding depletion does not take place. The concept is interesting but is a natural consequence of previous work on Nd UC sensitisation as well the properties of Nd in lasing technology. This concept has potential application perspective in multicolour STED super-resolution imaging using single combination of pump/depletion wavelengths, which has been demonstrated as well.

The article is well written in terms of text content, the state-of-the art is quite superficial, the figures are readable but sometimes chaotic and exclusionary in presenting only favourable features. There are some recurrences (e.g. Nd³⁺ energy diagram), which additionally differ from figure to figure. Therefore there are some drawbacks which make me hesitate about final opinion

and suggest the manuscript is not suitable in such form for acceptance.

1. Where is the $4F_{5/2}$ level of Nd^{3+} ? where is the cross-relaxation mechanism well known to affect the luminescent properties of Nd^{3+} ions? what motivated the authors to use 740 and now cw. 800 nm excitation wavelength? What motivated the authors to propose such combination of concentrations of ions (Nd^{3+}) or co-doping all ions homogeneously into nanoparticle (without displacing them in the core and shell)? These questions are important having in mind a complicated nature of energy transfer processes in multiple lanthanide co-doped systems, owing to effects such as cross-relaxation, phonon assisted transitions/ESA or back energy transfer. There are hints in SI, but in my opinion this is far too superficial. There are many more simplifications /superficial information, which do not build trust about reliability of analysis. Nd energy scheme has different level names or different levels, some levels are missing (for example CR) etc. Why ESA is disregarded under 740 or 1064 and phonon assisted transitions are dis-allowed in the interpretations? Why cannot the authors simplify the things and show energy graphs of Nd in single graph but in many different and sometimes incorrect versions (fig.S6-S8 + fig.2b (incorrect in some aspects))

2. both 740 and 1064 nm wavelength (of kW - MW/cm² pump densities) should contribute to excited state absorption, which seems is not considered in the discussion but can most probably be seen on the spectra (the transitions are not ascribed to adequate transitions). Moreover, the Ln^{3+} to Nd^{3+} ions "back" energy transfer has not been considered, which I believe may be a serious issue. I have to add I came to the same concept of using Nd^{3+} sensitizer as the entry point for multicolor up-conversion in 2013, but at that time we got very low (3%) depletion depth which we ascribed to the fact the excitation/depletion beam was populating Er directly through GSA or ESA, and was actually destroying the effect. We gave up at those times. I have a positive attitude to this work, but all these facts and my previous bad experiences makes me think the evidences presented by the authors are interesting but not sufficient or are not sufficiently well discussed.

a) I think the 1064 nm depletion beam power dependent full spectra must be shown and discussed and fig.3g is far too simplified. Fig.S15 show what I am asking for but this should be presented in main article to illustrate this is actually quite tricky to decode depletion efficiency when the corresponding emission band is localised on the spectral shoulder for the other emitting ions.

b) fig.2 and 3. are quite chaotically organised

c) because of cascade nature of the process the information about kinetics of the energy transfers and time required to achieve steady-state conditions is required to be studied and discussed. the "immediate" does not exhaust the topic.

3. "we achieve emission depletion for a wide range of emitters" is actually not correct, because this is the sensitization mechanism which is 'depleted' and not the emission of wide range of emitters - it doesn't come to emitting state population in 740+1064 combined together. Moreover, the issues explained in the abstract are not fully addressed - MW/cm² depletion intensity, which is high anyway, is still required. Moreover, what "multi-imaging nanoscopy" means?

4. The mechanism is clear for me and any specialist in Ln^{3+} spectroscopy, but for more general public, more details (fig.1) will definitely be beneficial for the readers.

5. there is no such thing as "long living transition"

6. "We also found that the optical depletion of the 655 nm emission from the $4F_{9/2}$ state is more efficient than that of the 540 nm emission from the $4S_{3/2}$ state, which was attributed to depletion-beam-driven excited state absorption (ESA, $4F_{9/2} \rightarrow 2H_{9/2}$)." indeed, the depletion efficiency should be the same for 540 and 650 nm emission if this is the same mechanism of Nd sensitisation undergoing depletion. But I am not sure I understand the explanation on ESA - if ESA occurs, the $4F_{9/2} \rightarrow 2H_{9/2}$ emission should be less prone to depletion, isn't it?

7. "Note that this STExD method can also circumvent the constraints of the emerging up-conversion super-resolution techniques 17,18" - which are? Above that, I also think the authors should be more modest in their conclusions about their work against state of the art, and also discuss the drawbacks of their work such as poor signal to noise or background ratio (which is not discussed at all), large size (as compared to organic dyes) of the luminescent probes which get

even larger when trying to passivate the surface of NPs, the dispersion between excitation and depletion may still cause issues with the two beams overlap, as well as fair comparison with other super-resolution techniques such as stochastic methods etc.

some typos should be carefully checked, e.g. "depletionn" etc. The article is also not equally good in terms of clarity/focus/level of details along its whole length.

Title: Achieving low-power, single-wavelength-pair nanoscopy with NIR-II CW laser for multi-chromatic probes

Point-by-Point Response to Reviewers

Dear Reviewers,

Thank you very much for your careful consideration and valuable comments on our manuscript, which have helped us significantly improve the manuscript. We have made every effort to address all the concerns raised by the reviewers. In response to the reviews, we made numerous changes to the manuscript in an attempt to enhance its clarity and readability. In the following, we provide a point-by-point response to the comments, together with the corresponding changes in the manuscript. As below, the reviewers' comments are written in **black** and our responses to them in **blue**. The important amendments or changes to the manuscript are given after the response in **red**.

Referee #1 (Remarks to the Author):

The diffraction limit problem has long hampered advances in microscopy towards better resolution and therefore detection of nanoscopic objects. One technique to overcome this limit is stimulated emission depletion (STED) microscopy. Since the invention of its principle in 1986 by Russian scientist Victor Okhonin, STED microscopy has however seen a slow development in view of the high-performing optical tools it needs, particularly high-power, ultrafast mode-locked lasers. Its first experimental demonstration was only made in 1999 by Stefan Hell who received the Nobel prize for it in 2014. In 2017, introduction of lanthanide-based upconversion nanoparticles (UCNPs) allowed achieving optical resolution of 28 nm at low-power excitation. Photo-switching in lanthanide nanoparticles is unfortunately rather limited and difficult to tune.

Here the authors bring a new dimension to the solution of the problem by proposing a depletion mechanism that works in the second biological window (NIR-II), a spectral range particularly useful for imaging. The idea is to transpose the depletion mechanism from the emitter to the sensitizer; in this way (STExD), subsequent emission depletion can be achieved for different emitters by using just one depletion wavelength. Here Nd (III) is used as sensitizer and after verifying emission depletion in an all-Nd (III) system, the authors successfully demonstrate its application to Nd/Yb/Ln nanoparticles (Ln = Er, Ho, Tm), Nd/Ln nanoparticles (Ln = Pr, Eu), and Gd/Tb@Gd/Nd/Yb/Tm nanoparticles. A common feature of these systems is the low saturation intensity (in the range 24-160 kW/cm²) and large depletion efficiency (the range 80-99%), overcoming by large the performances of other previously proposed systems.

As a proof-of-concept, the authors have built a nanoscope based on the STExD principle. Super-resolution images of nanoparticles have resolution down to 34 nm and immunolabelled subcellular filaments of HeLa cells could be revealed with resolution < 100 nm.

Generally speaking I find the demonstration rather convincing and the paper well and clearly written with quite didactic illustrations. The innovation presented here has great potential for diffraction-unlimited imaging both in physical and biological sciences.

Response: We would like to express our sincere appreciation to the reviewer for his/her effort to review this manuscript and his/her very positive evaluation and recommendation on our work. As suggested by the reviewer, in the revised “Introduction” we cited the literature about the invention of STED principle in 1986 by Victor Okhonin, and modified the progresses of STED super-resolution microscopy with upconversion nanoparticles.

Changes made in the revised manuscript:

Reference added (Page 2, introduction): “Super-resolution fluorescence microscopy encompasses a variety of techniques that can break the diffraction limit, and are of utmost importance for studies in life science and beyond¹⁻⁷. As a typical super-resolution technique, stimulated emission depletion (STED) microscopy provides diffraction-unlimited imaging resolution while preserving the merits of confocal fluorescence microscopy, such as optical sectioning, molecular specificity and fast imaging without the need for mathematical data postprocessing^{1-3,8-10}.”

1 Okhonin, V. Method of investigating specimen microstructure. *Patent SU*, 1374992 (1986).

Sentences modified (Page 2, introduction): “Lanthanide-doped upconversion nanoparticles have become an important group of luminophores for diverse applications due to their unique excitation and emission properties, and have been employed for STED nanoscopy featuring low-power NIR laser excitation and ultrahigh photostability¹⁶⁻²¹. However, photo-switching in lanthanide nanoparticles is unfortunately rather limited and difficult to tune, and the depletion laser needs to directly act on the emitting activators for emissive-state depopulation, which is essentially the same as in traditional STED implemented with organic dyes and quantum dots^{17,18}.”

16 Wu, R. *et al.* Optical depletion mechanism of upconverting luminescence and its potential for multi-photon STED-like microscopy. *Opt. Express* **23**, 32401-32412 (2015).

17 Liu, Y. *et al.* Amplified stimulated emission in upconversion nanoparticles for super-resolution nanoscopy. *Nature* **543**, 229-233 (2017).

18 Zhan, Q. *et al.* Achieving high-efficiency emission depletion nanoscopy by employing cross relaxation in upconversion nanoparticles. *Nat. Commun.* **8**, 1058 (2017).

Technical remarks

- I regret that elemental analysis of the nanoparticles (e.g. by EDS) is not given, particularly with respect to the doping content of the Ln ions

Response: We thank the reviewer for his/her suggestion. As requested by the reviewer, we added new experiments of elemental analysis. As shown in Fig. R1a, b, we firstly did EDS

(Energy Disperse Spectroscopy) analysis for the nanoparticle samples NaYF₄:Nd (3%) and Nd/Yb/Er (3/1/5%), confirming that all the lanthanide dopants were successfully doped into the particles as expected. To quantitatively characterize the concentrations of different dopants, we further made analysis using ICP (Inductive Coupled Plasma) emission spectrometer for most of the samples studied in this work, e.g., NaYF₄:Nd (3%), Nd/Yb/Er (3/1/5%), Nd/Yb/Ho (3/0.5/2%) , Nd/Pr (3/0.5%), Nd/Eu (3/15%), Nd/Yb/Tm (3/1/0.2%) and Nd/Ce (3/20%) nanoparticles. These analysis on element compositions show that the actual ion concentrations are basically in line with the experimental design (**Fig. R1c**).

Changes made in the revised manuscript:

Figure added (**Fig. R1**, supplementary Figure 4 in the revised manuscript): We added the analysis of EDS and ICP of the as-prepared nanoparticles in the revised Supplementary Information to quantitatively characterize the concentrations of different dopants.

Figure R1 (supplementary Figure 4 in the revised manuscript) The EDS and ICP analysis results of the as-prepared nanoparticles in this study. a, b The EDS spectrum of the as-prepared NaYF₄:Nd (3%) and Nd/Yb/Er (3/1/5%) nanoparticles. **c** ICP analysis of the as-prepared NaYF₄:Nd (3%), Nd/Yb/Er (3/1/5%), Nd/Yb/Ho (3/0.5/2%), Nd/Pr (3/0.5%), Nd/Eu (3/15%), Nd/Yb/Tm (3/1/0.2%) and Nd/Ce (3/20%) nanoparticles.

- Quantitative data (depletion efficiency, saturation intensity, lifetimes, a.s.o) should be associated with their experimental uncertainties.

Response: We thank the reviewer for his/her suggestion. As requested by the reviewer, we updated the quantitative data of depletion efficiency, depletion saturation intensity, lifetimes and doping concentration. Now in the revised manuscript, all the quantitative experimental data (depletion efficiency, depletion saturation intensity, lifetimes, nanoparticles size, doping concentration, a.s.o) have already been associated with their experimental uncertainties (see the updated **Fig. 2c, e, Fig. 3c** in the revised manuscript, **Figs. S3-6, S10-12, Tables S2 and S3** in

the revised Supplementary Information).

Changes made in the revised manuscript:

Please see the updated Figures and Tables in the revised manuscript and the revised SI.

Table added (Table R1, supplementary Table 3 in the revised Supplementary Information): We added the measured depletion efficiencies of different emission bands in the revised Supplementary Information to analyze the energy transfer process of excitation and emission inhibition of different Nd-sensitizing nanoparticles.

Table R1 (supplementary Table 3 in the revised manuscript). Measured depletion efficiencies of different emission bands of different Nd-sensitizing nanoparticles.

Sample	λ_{em} and transition	Depletion efficiency
NaYF ₄ :Nd (3%)	900 nm (⁴ F _{3/2} → ⁴ I _{9/2})	87.1 ± 0.6%
	867 nm (⁴ F _{3/2} → ⁴ I _{9/2})	87.7 ± 0.4%
	660 nm (² G _{9/2} → ⁴ I _{15/2})	97.0 ± 0.3%
	588 nm (² P _{1/2} → ⁴ I _{15/2})	96.4 ± 0.4%
	523 nm (⁴ G _{9/2} → ⁴ I _{9/2})	96.1 ± 0.4%
	450 nm (⁴ D _{3/2} → ⁴ I _{15/2})	99.3 ± 0.3%
	415 nm (⁴ P _{3/2} → ⁴ I _{11/2})	97.8 ± 0.5%
NaYF ₄ :Nd/Yb/Er (3/1/5%)	655 nm (⁴ F _{9/2} → ⁴ I _{15/2})	90.3 ± 0.4%
	540 nm (⁴ S _{3/2} → ⁴ I _{15/2})	80.7 ± 0.4%
	525 nm (² H _{11/2} → ⁴ I _{15/2})	74.3 ± 0.8%
	506 nm (⁴ G _{11/2} → ⁴ I _{13/2})	78.9 ± 0.6%
	472 nm (² K _{15/2} → ⁴ I _{13/2})	46.9 ± 0.8%
	410 nm (² H _{9/2} → ⁴ I _{15/2})	79.0 ± 0.7%
NaYF ₄ :Nd/Yb/Ho (3/0.5/2%)	645 nm (⁵ F ₅ → ⁵ I ₈)	82.2 ± 0.3%
	540 nm (⁵ F ₄ → ⁵ I ₈)	77.3 ± 0.6%
	488 nm (⁵ F ₃ → ⁵ I ₈)	70.0 ± 0.4%
NaYF ₄ :Nd/Pr (3/0.5%)	642 nm (³ P ₀ → ³ H ₆)	46.2 ± 0.6%
	610 nm (¹ D ₂ → ³ H ₄)	35.5 ± 0.7%
	485 nm (³ P ₀ → ³ H ₄)	35.0 ± 0.9%
NaYF ₄ :Nd/Eu (3/15%)	615 nm (⁵ D ₀ → ⁷ F ₁)	88.1 ± 1.7%
NaYF ₄ :Nd/Yb/Tm (3/1/0.2%)	475 nm (¹ G ₄ → ³ H ₆)	55.1 ± 1.3%
NaYF ₄ :Gd/Tb@Gd/Nd/Yb/Tm (50/15@50/3/5/1%)	620 nm (⁵ D ₄ → ⁵ F ₃)	48.8 ± 0.9%
	545 nm (⁵ D ₄ → ⁵ F ₅)	58.1 ± 0.7%

In closing, we would like to thank the reviewer again for taking the time to review our manuscript. We hope that the reviewer will find this revised manuscript substantially improved.

Reviewer #2 (Remarks to the Author):

The review of an article "Achieving low-power, single-wavelength-pair nanoscopy with NIR-II CW laser for multi-chromatic probes" by Xin Guo et al.

The article presents an interesting concept of using single Nd^{3+} sensitizer ions as the starting point to $\text{Yb}^{3+}\text{-Ln}^{3+}$ ($\text{Ln}=\text{Er, Ho, Tm, Pr, Tb, Eu}^{3+}$ ions) upconversion. The Nd^{3+} ions, were shown to be excited under ca.800nm and undergo significant depletion under 1064 nm. The luminescence from such co-doped NPs occurs from the co-doping ions (the Ln^{3+} listed above) under the condition the proceeding depletion does not take place. The concept is interesting but is a natural consequence of previous work on Nd UC sensitization as well the properties of Nd in lasing technology. This concept has potential application perspective in multicolour STED super-resolution imaging using single combination of pump/depletion wavelengths, which has been demonstrated as well.

Response: We would like to express our sincere appreciation to the reviewer for his/her effort to review this manuscript and for his/her affirmation of the meaning and potential of our work.

The article is well written in terms of text content, the state-of-the art is quite superficial, the figures are readable but sometimes chaotic and exclusionary in presenting only favourable features. There are some recurrences (e.g. Nd^{3+} energy diagram), which additionally differ from figure to figure. Therefore there are some drawbacks which make me hesitate about final opinion and suggest the manuscript is not suitable in such form for acceptance.

Response: We also thank the reviewer for his/her comments and suggestions, according to which we have systematically revised the manuscript to improve its quality and readability. About the state-of-the art, as suggested by the reviewers, in the revised "Introduction" we modified the sentences to analyze the present status of technology, and cited more critical literatures about the progresses of STED microscopy and upconversion STED techniques.

In addition, we re-organized the **Fig. 2** and **Fig.3** to enhance the clarity and readability. We updated the energy level diagrams and added more details of energy transfer processes, e.g., cross-relaxation (CR) and excited state absorption (ESA) processes, to analyze the energy transfer mechanism of Nd-sensitizing systems (**Fig. 2a, Fig. 3a and Fig. S15** in the revised manuscript). We carried out new experiments to analyze the brightness, the depletion efficiency and saturation intensity of samples with different Nd^{3+} concentrations (**Fig. 2e and Fig. S10** in the revised manuscript) and the effects of different excitation wavelength at 740 nm and 800 nm (**Fig. S5** in the revised manuscript). In addition, we removed duplicate energy level diagrams (**Fig. 2a and Fig. S6** in the original manuscript).

See below for the details of our point-to-point response to each specific comment.

Changes made in the revised manuscript:

Reference added (**Page 2, introduction**): "Super-resolution fluorescence microscopy encompasses a variety of techniques that can break the diffraction limit, and are of utmost importance for studies in life science and beyond¹⁻⁷. As a typical super-resolution technique,

stimulated emission depletion (STED) microscopy provides diffraction-unlimited imaging resolution while preserving the merits of confocal fluorescence microscopy, such as optical sectioning, molecular specificity and fast imaging without the need for mathematical data postprocessing^{1-3,8-10}.”

1 Okhonin, V. Method of investigating specimen microstructure. *Patent SU*, 1374992 (1986).

Sentences modified (Page 2, introduction): “In addition, high-precision overlapping of multiple pairs of lasers is also typically required for multi-chromatic probes and for multi-color super-resolution imaging to match their different energy states, which results in high complexity, high cost and low usability (Fig. 1a)^{13,14}. Meanwhile, the spectral overlap among the utilized dyes (typically with broad emission spectra) would also lead to cross-talk between channels¹⁵, and thereby compromises its utility and impact (Fig. 1a).

Lanthanide-doped upconversion nanoparticles have become an important group of luminophores for diverse applications due to their unique excitation and emission properties, and have been employed for STED nanoscopy featuring low-power NIR laser excitation and ultrahigh photostability¹⁶⁻²¹. However, photo-switching in lanthanide nanoparticles is unfortunately rather limited and difficult to tune, and the depletion laser needs to directly act on the emitting activators for emissive-state depopulation, which is essentially the same as in traditional STED implemented with organic dyes and quantum dots^{17,18}.”

16 Wu, R. *et al.* Optical depletion mechanism of upconverting luminescence and its potential for multi-photon STED-like microscopy. *Opt. Express* **23**, 32401-32412 (2015).

17 Liu, Y. *et al.* Amplified stimulated emission in upconversion nanoparticles for super-resolution nanoscopy. *Nature* **543**, 229-233 (2017).

18 Zhan, Q. *et al.* Achieving high-efficiency emission depletion nanoscopy by employing cross relaxation in upconversion nanoparticles. *Nat. Commun.* **8**, 1058 (2017).

1. Where is the $^4F_{5/2}$ level of Nd^{3+} ? Where is the cross-relaxation mechanism well known to affect the luminescent properties of Nd^{3+} ions? what motivated the authors to use 740 and no cw. 800 nm excitation wavelength? What motivated the authors to propose such combination of concentrations of ions (Nd3%) or do-doping all ions homogeneously into nanoparticle (without displacing them in the core and shell)? This questions are important having in mind a complicated nature or energy transfer processes in multiple lanthanide co-doped systems, owing to effects such as cross-relaxation, phono assisted transitions/ESA or back energy transfer. There are hints in SI, but in my opinion this is far too superficial. There are many more simplifications /superficial information, which do not build trust about reliability of analysis. Nd energy scheme has different level names or different levels, some levels are missing (for example CR) etc. Why ESA is disregarded under 740 or 1064 and phonon assisted transitions are dis-allowed in the interpretations? Why cannot the authors simplify the things and show energy graphs of Nd in single graph but in many different and sometimes incorrect versions

(fig. S6-S8 + fig. 2b (incorrect in some aspects))

Response: We thank the reviewer for his/her suggestions and comments. Our thoughts and responses are given below point to point.

- 1) In the original manuscript, to make the diagram simple we removed some unimportant energy levels of Nd^{3+} ion that have negligible effect on the mechanism. As suggested by the reviewer, in this revision we added those energy levels and updated the diagram with complete energy levels, including ${}^4\text{F}_{5/2}$, ${}^4\text{F}_{9/2}$, ${}^2\text{H}_{11/2}$, ${}^2\text{G}_{9/2}$ and ${}^2\text{D}_{5/2}$ (**Fig. R2a**). We removed duplicate energy level diagrams (**Fig. 2a and Fig. S6** in the original manuscript).
- 2) To highlight the mechanism of cascade amplified depletion, we separated sensitizer and activator to describe this process with a widely used term of energy transfer upconversion (ETU, *IEEE J Quantum Electron* **1999**, **35**, 647; *Opt. Express* **2018**, **26**, 6478), including ${}^4\text{F}_{3/2} + {}^4\text{I}_{9/2} \rightarrow {}^4\text{I}_{9/2} + {}^4\text{F}_{3/2}$ (ETU1), ${}^4\text{F}_{3/2} + {}^4\text{F}_{3/2} \rightarrow {}^4\text{I}_{9/2} + {}^2\text{P}_{1/2}$ (ETU2), and ${}^4\text{F}_{3/2} + {}^4\text{G}_{5/2} \rightarrow {}^4\text{I}_{9/2} + {}^4\text{D}_{5/2}$ (ETU3), which also can be described as CR processes (**Fig. R2a**).
- 3) We added a new CR process of ${}^4\text{G}_{9/2} + {}^4\text{I}_{9/2} \rightarrow {}^4\text{G}_{5/2} + {}^4\text{I}_{11/2}$ in Nd^{3+} (**Fig. R2a**), which leads to lower depletion efficiency and higher saturation intensity in high concentration Nd-nanoparticles (**Fig. R2e**). The weakened emission bands of 523 nm (${}^4\text{G}_{9/2} \rightarrow {}^4\text{I}_{9/2}$) and 533 nm (${}^4\text{G}_{7/2} \rightarrow {}^4\text{I}_{9/2}$) and the enhanced emission band of 573 nm (${}^4\text{G}_{5/2} \rightarrow {}^4\text{I}_{9/2}$) in $\text{NaYF}_4:\text{Nd}$ (10%) nanoparticles compared with Nd (3%) can be also attributed to this CR process (**Fig. R3a, b**). The CR process of ${}^4\text{F}_{3/2} + {}^4\text{I}_{9/2} \rightarrow {}^4\text{I}_{15/2} + {}^4\text{I}_{15/2}$ already existed in the original version.
- 4) We added an ESA processes ${}^4\text{G}_{5/2} \rightarrow {}^2\text{P}_{3/2}$ of 1064 nm in Nd^{3+} with low absorption coefficient (**Fig. R2a**), which leads to a slightly lower depletion efficiency ($97.8 \pm 0.5\%$) for 415 nm emission compared with that ($99.3 \pm 0.3\%$) of 450 nm emission (**Fig. R2b**). If its ESA process is strong, the depletion efficiency of 415 nm emission should be significantly reduced.
- 5) The 740 nm beam matches the energy gap of ${}^4\text{F}_{5/2} \rightarrow {}^2\text{P}_{3/2}$, but the rapid non-radially relaxation of ${}^4\text{F}_{7/2} \rightarrow {}^4\text{F}_{5/2} \rightarrow {}^4\text{F}_{3/2}$ causes the population of ${}^4\text{F}_{5/2}$ level to be almost empty, which is not enough to support the ESA process. Moreover, if the ESA is strong, the depletion efficiency of multi-photon emissions should not be significantly higher than one-photon emissions. Because only ETU emissions have the effect of depletion amplification. The measured decreased depletion saturation intensities of multi-photon emissions compared with one-photon emission also confirms this result (from 116.2 ± 1.3 to 23.8 ± 0.4 kW cm^{-2} , **Fig. R2c**).
- 6) The reason why we chose the concentration of 3% in Nd-nanoparticles: In Nd^{3+} -doped nanoparticles, high concentration ($> 3\%$) will increase the rate of CR, such as ${}^4\text{F}_{7/2} + {}^4\text{I}_{9/2} \rightarrow {}^4\text{F}_{3/2} + {}^4\text{I}_{11/2}$, which will weaken the effect of population inversion, and then decrease the depletion efficiency and elevate the required saturation intensity (I_{sat}) (**Fig. R2e**). The I_{sat} of 588 nm emission rises by 50% from 42.4 ± 0.7 to 61.1 ± 1.2 kW cm^{-2} when the concentration increases from 3% to 4%. Meanwhile, the Nd^{3+} ion is not only the sensitizer

for absorbing the excitation energy, but also the activator for emitting. Low concentration ($\leq 2\%$) will limit the emission brightness (**Fig. R3c, d**). We chose the concentration of 3% after trading off the emission intensity and the saturation intensity I_{sat} .

- 7) The reason why we used 740 nm excitation: First, 740 nm band is also a commonly used excitation band for Nd^{3+} (*Phys. Rev. B* **2010**, **81**, 035107; *Opt. Commun.* **2011**, **284**, 603). The absorption cross-section of Nd^{3+} is very large at both 740 nm and 800 nm band (**Fig. R4a**), and these two excitation bands do not make difference to the depletion efficiencies of Nd^{3+} and Ho^{3+} emissions (**Fig. R2b and R4b, d**). Second, the 800 nm laser can directly excite Er^{3+} (not via Nd^{3+}), which would reduce the depletion efficiency (*J. Appl. Phys.* **1998**, **85**, 29; *Annu Rev Phys Chem.* **2015**, **17**, 11481). We added a new experiment. Under the co-irradiation of 800 nm and 1064 nm beam, the depletion efficiencies of 655 nm emission band from $\text{NaYF}_4:\text{Nd}/\text{Yb}/\text{Er}$ (3/1/5%) nanoparticles is only $38.9 \pm 1.7\%$ (**Fig. R4c**).
- 8) The reason why we doped all ions homogeneously into one core: We agree the reviewer that the phenomenon of back energy transfer from Er^{3+} to Nd^{3+} (${}^4\text{I}_{11/2} + {}^4\text{I}_{9/2} \rightarrow {}^4\text{I}_{13/2} + {}^4\text{I}_{13/2}$ and ${}^4\text{I}_{13/2} + {}^4\text{I}_{9/2} \rightarrow {}^4\text{I}_{15/2} + {}^4\text{I}_{15/2}$) exists in $\text{Nd}/\text{Yb}/\text{Er}$ system (*Adv. Mater.* **2014**, **26**, 2831), but in the low-concentration doping system (3% Nd^{3+} in our study, traditionally 20% Nd^{3+}) the impact of this problem is not significant. Another important consideration is that the epitaxially grown core-shell structured nanoparticles have much larger volume, which is undesirable for immunolabeling and super-resolution biological imaging. We added new supplementary experiments, and the results show that the 655 nm emission (used for STExD imaging) intensity of the $\text{NaYF}_4:\text{Er}/\text{Yb}@/\text{Yb}/\text{Nd}$ (5/1@1/3%) nanoparticles was enhanced by 34% compared with $\text{NaYF}_4:\text{Nd}/\text{Yb}/\text{Er}$ (3/1/5%) nanoparticles, while the particles diameter increased from ca. 8.4 nm to 18.9 nm (8.6 nm core, shell thickness 5.1 nm, volume: 310 nm^3 versus 3535 nm^3) (**Fig. R5**). Anyway, it is worth noting that the prepared small particles ($\text{Nd}/\text{Yb}/\text{Er}$, $\text{Nd}/\text{Yb}/\text{Ho}$) in our study can be used for single-nanoparticles super-resolution imaging (Figure 4 in the manuscript), which means they are capable for future biological applications.

Changes made in the revised manuscript:

Please see Figures R2-R5 (Figures 2 in the revised manuscript and Supplementary Figures 5, 10 in the revised Supplementary Information).

Accordingly, the rate equation simulations have been modified and new calculation results were updated in the revised Supplementary Information (see Discussions 1, and Supplementary Figures 1, 8, 9).

Sentences modified (Page 3, results): “Guided by the results from numerical simulations, a lab-made microscopy system coupled with a NIR continuous-wave (CW) excitation beam at 740 nm²⁴ and a NIR-II CW depletion beam at 1064 nm was built to perform spectroscopic and imaging studies (**Fig. S2**).”

- 24 Meijer, J.-M., Aarts, L., van der Ende, B. M., Vlugt, T. J. H. & Meijerink, A. Downconversion for solar cells in $\text{YF}_3:\text{Nd}^{3+}, \text{Yb}^{3+}$. *Phys. Rev. B* **81**, 035107 (2010).

Sentences modified (Page 5, results): “When increasing the doping concentration (1~10%), the high depletion efficiency was hardly affected and the saturation intensities were slightly elevated (Fig. 2e), which could be ascribed to the fact that the enhanced cross-relaxation (CR) process of ${}^4G_{9/2} + {}^4I_{9/2} \rightarrow {}^4G_{5/2} + {}^4I_{11/2}$ would compete with the stimulated emission process (Fig. 2e and Fig S10a, b). In addition, the brightness of particles first increases and then saturate with increasing doping concentration, governed by the laser absorption and the CR process of ${}^4F_{3/2} + {}^4I_{9/2} \rightarrow {}^4I_{15/2} + {}^4I_{15/2}$ ^{27,28} (Fig. S10c, d).”

27 Camargo, A. S. S. *et al.* Effect of Nd³⁺ concentration quenching in highly doped lead lanthanum zirconate titanate transparent ferroelectric ceramics. *J. Appl. Phys.* **101**, 053111 (2007).

28 Reddy Prasad, V., Seshadri, M., Babu, S. & Ratnakaram, Y. C. Concentration-dependent studies of Nd³⁺-doped zinc phosphate glasses for NIR photoluminescence at 1.05 μm . *Luminescence* **32**, 443-451 (2017).

Sentences added (Page 4, results): “The slight discrepancy of depletion efficiency between 415 nm and 450 nm emissions may be attributed to the excited state absorption (ESA) process of ${}^4G_{5/2} \rightarrow {}^2P_{3/2}$ driven by the depletion beam.”

Figure R2 (Figure 2 in the revised manuscript) Cascade amplified depletion of STExD in NaYF₄:Nd nanoparticles. **a** Energy level diagram of Nd-upconversion system illustrating the emission inhibition process with the 1064 nm beam. **b** Full-spectrum emission inhibition and the depletion efficiencies of different emissions of NaYF₄:Nd (3%) nanoparticles under 740 nm and 1064 nm co-irradiation. **c** Emission intensity of one-photon (867 nm), two-photon (588 nm) and three-photon (450 nm) emissions from NaYF₄:Nd (3%) nanoparticles versus depletion laser intensity ($I_{740} = 78.3 \text{ kW cm}^{-2}$, $I_{1064} = 0-2.5 \text{ MW cm}^{-2}$). Inset: the measured I_{sat} were 116.2 ± 1.3 , 42.4 ± 0.7 , and $23.8 \pm 0.4 \text{ kW cm}^{-2}$ for 867 nm, 588 nm and 450 nm emissions, respectively. **d**

Comparison between calculated depletion efficiencies (emission intensity ratio, $(I_{740} - I_{740\&1064})/I_{740}$) of one-, two- and three-photon emissions. **e** The measured depletion efficiency and saturation intensity (I_{sat} , 50% emission off) of 588 nm emission versus the Nd^{3+} concentration. $\text{NaYF}_4:\text{Nd}$ (x%) nanoparticles with $x = 1, 2, 3, 4, 5, 8, 10$. Error bars are defined as the standard deviation of $n = 3$ independent measurements.

Figure added (Figure R3, supplementary Figure 10 in the revised Supplementary Information): We added the measured emission spectrums of $\text{NaYF}_4:\text{Nd}$ nanoparticles with different doping concentration in the revised Supplementary Information to analyze the emission brightness and the process of cross-relaxation.

Figure R3 (supplementary Figure 10 in the revised manuscript). **a** Normalized emission spectrums show the weakened emission bands of 523 nm (${}^4G_{9/2} \rightarrow {}^4I_{9/2}$) and 533 nm (${}^4G_{7/2} \rightarrow {}^4I_{9/2}$) and the enhanced emission band of 573 nm (${}^4G_{5/2} \rightarrow {}^4I_{9/2}$) in $\text{NaYF}_4:\text{Nd}$ (10%) nanoparticles compared with Nd (3%). **b** The associated energy level diagram of the CR process: ${}^4G_{9/2} + {}^4I_{9/2} \rightarrow {}^4G_{5/2} + {}^4I_{11/2}$. **c** Emission spectrums of $\text{NaYF}_4:\text{Nd}$ (x%) nanoparticles with $x = 1, 2, 3, 5, 8, 10$. **d** Measured emission intensities of 588 nm emission in (c). Error bars are defined as the standard deviation of $n = 3$ independent measurements. $I_{740} = 78.3 \text{ kW cm}^{-2}$.

Figure R4 (supplementary Figure 5 in the revised manuscript). **a** The absorption spectrum of Nd^{3+} ions shows large absorption cross-section at 740 nm and 800 nm band, and almost no absorption at 1064 nm⁶. **b-d** Emission inhibition of $\text{NaYF}_4:\text{Nd}$ (3%), $\text{Nd}/\text{Yb}/\text{Er}$ (3/1/5%) and $\text{Nd}/\text{Yb}/\text{Ho}$ (3/0.5/2%) nanoparticles with the co-irradiation of 800 nm and 1064 nm beam. The measured depletion efficiencies are $96.3 \pm 0.3\%$ (Nd-588 nm), $38.9 \pm 1.7\%$ (Er-655 nm), $80.2 \pm 0.6\%$ (Ho-645 nm). Uncertainties are defined as the standard deviation of $n = 3$ independent measurements.

Figure R5. Emission spectra of single core $\text{NaYF}_4:\text{Nd}/\text{Yb}/\text{Er}$ (3/1/5%) and core-shell structured $\text{NaYF}_4:\text{Yb}/\text{Er}@\text{Yb}/\text{Nd}$ (1/5@1/3%) nanoparticles under the laser excitation, $I_{740} = 78.3 \text{ kW cm}^{-2}$.

2. both 740 and 1064 nm wavelength (of kW - MW/cm^2 pump densities) should contribute to excited state absorption, which seems is not considered in the discussion but can most probably be seen on the spectra (the transitions are not ascribed to adequate transitions). Moreover, the

Ln³⁺ to Nd³⁺ ions "back" energy transfer has not been considered, which I believe may be a serious issue. I have to add I came to the same concept of using Nd³⁺ sensitizer as the entry point for multicolor up-conversion in 2013, but at that time we got very low (3%) depletion depth which we ascribed to the fact the excitation/depletion beam was populating Er directly through GSA or ESA, and was actually destroying the effect. We gave up at those times. I have a positive attitude to this work, but all these facts and my previous bad experiences makes me think the evidences presented by the authors are interesting but not sufficient or are not sufficiently well discussed.

Response: We thank the reviewer for his/her comments.

In Nd/Yb/Er system, we added new energy levels of ⁴F_{5/2}, ⁴F_{3/2}, ⁴G_{11/2}, ⁴G_{9/2} and ²K_{5/2} and two emission bands of 472 nm (²K_{15/2} → ⁴I_{13/2}) and 506 nm (⁴G_{11/2} → ⁴I_{13/2}) by the detailed analysis of the emission spectrum (**Fig. R6a**). We added a new ESA process of 1064 nm: ⁴S_{3/2} → ²K_{15/2}, which reduced the depletion efficiency of 472 nm (46.9%) emission band compared with the 540 nm (80.7 ± 0.4%) and 655 nm (90.3% ± 0.4) emission bands (**Fig. R6b**). The ESA process of ⁴F_{9/2} → ²H_{9/2} existed in the original manuscript.

In Nd/Yb/Ho system, we added the ESA process of 740 nm: ⁵I₇ → ⁵F₄, and the ESA process of 1064 nm: ⁵I₅ → ⁵F₃. The ESA process of 1064 nm: ⁵I₆ → ⁵F₄ existed in the original manuscript should be a phonon assisted transition due to the unmatched energy gap (**Fig. R6a**). The ESA process of 740 nm leads to lower depletion efficiency of Ho³⁺ emissions compared with the Nd/Yb/Er system, while the ESA processes of 1064 nm leads to further lower depletion efficiency of 488 nm (70.0 ± 0.4%) emission band (**Fig. R6d**).

We thank the reviewer very much for sharing their results from trial studies. As mentioned above, we got a similar result as the reviewer mentioned and experienced. The depletion efficiency of 655 nm emission is only 38.9 ± 1.7% in NaYF₄:Nd/Yb/Er (3/1/5%) nanoparticles under 800 and 1064 nm co-irradiation, which is mainly due to the efficient excitation effect of the 800 nm beam (**Fig. R4c**). Additionally, in high-doping nanoparticles (such as Nd/Yb 20/20%), we also got further decreased depletion efficiency of Er³⁺ emission (less than 10%), which is considered to be due to the undesired CR (⁴G_{9/2} + ⁴I_{9/2} → ⁴G_{5/2} + ⁴I_{11/2}) of Nd³⁺ and the phonon-assisted absorption of high power 1064 nm beam by Yb³⁺.

As mentioned above, we also discussed the influence of back energy transfer from Er³⁺ to Nd³⁺. In the low-concentration doping system, the impact of this problem is not significant (**Fig. R5**).

Changes made in the revised manuscript:

Please see Figure R6 (Figure 3 in the revised manuscript).

Accordingly, the rate equation simulations have been modified and new calculation results were updated in the revised Supplementary Information (see Discussions 3, 4 and Supplementary Figures 13, 14).

Sentences modified (Page 6, results): "As shown in **Fig. 3a**, upon excitation at 740 nm, the activator Er³⁺ obtained excitation energy from Yb³⁺ and Nd³⁺ via multistep energy transfer, displaying characteristic emission bands centered at 655 nm (⁴F_{9/2} → ⁴I_{15/2}), 540 nm (⁴S_{3/2} →

$^4I_{15/2}$), 525 nm ($^2H_{11/2} \rightarrow ^4I_{15/2}$), 506 nm ($^4G_{11/2} \rightarrow ^4I_{13/2}$), 472 nm ($^2K_{15/2} \rightarrow ^4I_{13/2}$) and 410 nm ($^2H_{9/2} \rightarrow ^4I_{15/2}$)³¹ (Fig. 3b and Fig. S12a), and the 588 nm emission of Nd³⁺ is also presented³².”

31 Yuan, M. *et al.* Exploiting the silent upconversion emissions from a single β -NaYF₄:Yb/Er microcrystal via saturated excitation. *J. Mater. Chem. C* **6**, 10226-10232 (2018).

32 Zhong, Y. *et al.* Elimination of Photon Quenching by a Transition Layer to Fabricate a Quenching-Shield Sandwich Structure for 800 nm Excited Upconversion Luminescence of Nd³⁺-Sensitized Nanoparticles. *Adv. Mater.* **26**, 2831-2837 (2014).

Sentences modified (Page 6, results): “When Er³⁺-activated nanoparticles were excited by the 740 nm laser (non-resonant with Er³⁺) instead of the 800 nm laser (resonant with Er³⁺, Fig. S5c), the depletion efficiency of the 655 nm emission was measured to be $90.3 \pm 0.4\%$ (Fig. 3c and Table S3), which by far exceeds the previously reported depletion efficiency of 27% in Er³⁺-activated nanoparticles¹⁶, with a saturation intensity of 160.9 ± 1.7 kW cm⁻². The higher depletion efficiency of the 655 nm emission than that of the emissions from higher states was attributed to the 1064-nm laser induced ESA processes of $^4F_{9/2} \rightarrow ^2H_{9/2}$ and $^4S_{3/2} \rightarrow ^2K_{15/2}$.”

Sentences added (Page 6, results): “The ESA processes of $^5I_7 \rightarrow ^5F_4$, $^5I_6 \rightarrow ^5F_4$ (phonon assisted) and $^5I_5 \rightarrow ^5F_3$ compromised the depletion efficiencies of Ho³⁺ emissions, i.e., 650 nm, 540 nm and 488 nm emissions, respectively.”

Figure R6 (Figure 3 in the revised manuscript) STExD strategy extended to various Nd-sensitized nanoparticles. **a** Optical emission inhibition mechanism in Nd/Yb/Er and Nd/Yb/Ho systems. **b** Full-spectrum emission inhibition of NaYF₄:Nd/Yb/Er (3/1/5%) nanoparticles under 740 nm and 1064 nm co-irradiation ($I_{740} = 78.3 \text{ kW cm}^{-2}$; $I_{1064} = 2.5 \text{ MW cm}^{-2}$). **c** Emission intensity at the 655 nm of Er³⁺ and the 645 nm of Ho³⁺ versus the depletion intensity ($I_{740} = 78.3 \text{ kW cm}^{-2}$; $I_{1064} = 0\text{-}2.5 \text{ MW cm}^{-2}$). The measured I_{sat} were 160.9 ± 1.7 (Er-655 nm) and 85.8 ± 0.7 (Ho-645 nm) kW cm⁻². Error bars are defined as the standard deviation of $n = 3$ independent measurements. **d-h** Full-spectrum emission inhibition of other nanoparticles NaYF₄:Nd/Yb/Ho (3/0.5/2%), Nd/Pr (3/0.5%), Nd/Eu (3/15%), Nd/Yb/Tm (3/1/0.2%) and Gd/Tb@Gd/Nd/Yb/Tm (50/15@50/3/5/1%) under 740 nm and 1064 nm co-irradiation.

a) I think the 1064 nm depletion beam power dependent full spectra must be shown and discussed and fig.3g is far too simplified. Fig.S15 show what I am asking for but this should be presented in main article to illustrate this is actually quite tricky to decode depletion efficiency when the corresponding emission band is localized on the spectral shoulder for the other emitting ions.

Response: We thank the reviewer for his/her comments.

As suggested by the reviewer, we have presented the full-spectrum emission inhibition of Pr-, Eu-, Tm- and Tb-nanoparticles in Figure 3 in the revised manuscript and reanalyzed their energy transfer pathways (**Fig. R6e-h and R7**). In Nd/Pr system, we added new ESA processes of 740 nm ($^3\text{F}_2 \rightarrow ^3\text{P}_0$) and 1064 nm ($^3\text{F}_2 \rightarrow ^1\text{D}_2$). In Nd/Yb/Tm system, we also added a new ESA process of 1064 nm ($^3\text{F}_4 \rightarrow ^3\text{F}_2$). These ESA processes would somehow reduce the emission depletion efficiencies of Pr³⁺, Tm³⁺ and Tb³⁺.

We also added the depletion beam power dependent full spectra of Pr-nanoparticles, which clearly shows the gradually emission inhibition with the increasing laser intensity of 1064 nm (**Fig. R8**).

We thank the reviewer for pointing out the issue of decoding the depletion efficiency of emission bands localized on the spectral shoulder for the other emitting ions. In order to decode the depletion efficiency more accurately, we adopted the spectrum-unmixing technique proposed in *Nat. Photonics* **2020**, **14**, 760-766 and tried to unmix the overlapping spectra of Er³⁺, Ho³⁺, Pr³⁺, and Tb³⁺ from Nd³⁺ emissions by normalizing the 588 nm emission band, as shown in **Fig. R6b, R6d, R6e and R6h**. The recalculated emission depletion efficiency of 655 nm, 525 nm and 410 nm in Er-nanoparticles are 90.3%, 74.3% and 79.0, respectively. The depletion efficiency of 645 nm in Ho-nanoparticles is 82.2%. In the Pr-nanoparticles, the emission depletion efficiency of 485 nm, 610 nm and 642 nm are 35.0%, 35.5% and 46.2%, respectively. In Tb-nanoparticles, the emission depletion efficiency of 545 nm and 620 nm bands are 58.1% and 48.8%, respectively.

Changes made in the revised manuscript:

Please see Figure R6-R8 and Table R1 (Figure 3 in the revised manuscript, supplementary

Figure 15 and supplementary Table 3 in the revised Supplementary Information).

Sentences added (Page 6, results): “We unmixed the emission spectrum of Er^{3+} from Nd^{3+} by normalizing the 588 nm emission band.”

Sentences added (Page 6, results): “The effect of emission inhibition was achieved for Pr^{3+} (642 nm, 610 nm and 485 nm emissions), Eu^{3+} (615 nm emission), Tm^{3+} (475 nm emission) and Tb^{3+} (620 and 545 nm emissions) (Fig. 3e-h and Table S3). The observed depletion efficiencies of the emissions from Pr^{3+} and Tm^{3+} were relatively low due to the ESA processes of 740 nm (${}^3\text{F}_2 \rightarrow {}^3\text{P}_0$ of Pr^{3+}) and 1064 nm (${}^3\text{F}_2 \rightarrow {}^1\text{D}_2$ of Pr^{3+} and ${}^3\text{F}_4 \rightarrow {}^3\text{F}_2$ of Tm^{3+}) (Fig. S15).”

Figure R7 (supplementary Figure 15 in the revised manuscript). Schematic energy diagram of (a) the Nd/Pr, (b) the Nd/Eu, (c) the Nd/Yb/Tm and Gd/Tb@Gd/Nd/Yb/Tm systems.

Figure R8. The depletion beam power dependent full spectra of $\text{NaYF}_4:\text{Nd}/\text{Pr}$ (3/0.5%)

nanoparticles ($I_{740} = 78.3 \text{ kW cm}^{-2}$; $I_{1064} = 0, 0.5, 1 \text{ and } 2.5 \text{ MW cm}^{-2}$).

Table added (Table R1, supplementary Table 3 in the revised Supplementary Information): We added the measured depletion efficiencies of different emission bands in the revised Supplementary Information to analyze the energy transfer process of excitation and emission inhibition of different Nd-sensitizing nanoparticles.

Table R1 (supplementary Table 3 in the revised manuscript). Measured depletion efficiencies of different emission bands of different Nd-sensitizing nanoparticles.

Sample	λ_{em} and transition	Depletion efficiency
NaYF ₄ :Nd (3%)	900 nm (⁴ F _{3/2} → ⁴ I _{9/2})	87.1 ± 0.6%
	867 nm (⁴ F _{3/2} → ⁴ I _{9/2})	87.7 ± 0.4%
	660 nm (² G _{9/2} → ⁴ I _{15/2})	97.0 ± 0.3%
	588 nm (² P _{1/2} → ⁴ I _{15/2})	96.4 ± 0.4%
	523 nm (⁴ G _{9/2} → ⁴ I _{9/2})	96.1 ± 0.4%
	450 nm (⁴ D _{3/2} → ⁴ I _{15/2})	99.3 ± 0.3%
	415 nm (⁴ P _{3/2} → ⁴ I _{11/2})	97.8 ± 0.5%
NaYF ₄ :Nd/Yb/Er (3/1/5%)	655 nm (⁴ F _{9/2} → ⁴ I _{15/2})	90.3 ± 0.4%
	540 nm (⁴ S _{3/2} → ⁴ I _{15/2})	80.7 ± 0.4%
	525 nm (² H _{11/2} → ⁴ I _{15/2})	74.3 ± 0.8%
	506 nm (⁴ G _{11/2} → ⁴ I _{13/2})	78.9 ± 0.6%
	472 nm (² K _{15/2} → ⁴ I _{13/2})	46.9 ± 0.8%
	410 nm (² H _{9/2} → ⁴ I _{15/2})	79.0 ± 0.7%
NaYF ₄ :Nd/Yb/Ho (3/0.5/2%)	645 nm (⁵ F ₅ → ⁵ I ₈)	82.2 ± 0.3%
	540 nm (⁵ F ₄ → ⁵ I ₈)	77.3 ± 0.6%
	488 nm (⁵ F ₃ → ⁵ I ₈)	70.0 ± 0.4%
NaYF ₄ :Nd/Pr (3/0.5%)	642 nm (³ P ₀ → ³ H ₆)	46.2 ± 0.6%
	610 nm (¹ D ₂ → ³ H ₄)	35.5 ± 0.7%
	485 nm (³ P ₀ → ³ H ₄)	35.0 ± 0.9%
NaYF ₄ :Nd/Eu (3/15%)	615 nm (⁵ D ₀ → ⁷ F ₁)	88.1 ± 1.7%
NaYF ₄ :Nd/Yb/Tm (3/1/0.2%)	475 nm (¹ G ₄ → ³ H ₆)	55.1 ± 1.3%
NaYF ₄ :Gd/Tb@Gd/Nd/Yb/Tm (50/15@50/3/5/1%)	620 nm (⁵ D ₄ → ⁵ F ₃)	48.8 ± 0.9%
	545 nm (⁵ D ₄ → ⁵ F ₅)	58.1 ± 0.7%

b) fig.2 and 3. are quite chaotically organized

Response: We thank the reviewer for pointing out the flaw of our figure presentation. In this revision, we have re-organized and modified Fig. 2 and Fig. 3 (**Fig. R2** and **R6**) as follows.

In **Fig. 2**, we updated the energy level diagram and combined the repeated diagram of energy level (**Fig. 2a** in the original manuscript) to supplementary Figure (**Fig. R9**). The depletion efficiencies of Nd³⁺ emission bands were combined with the emission inhibition spectrum (**Fig. R2b**). the schematic diagram and calculated depletion rates of multiphoton upconversion

system were combined to the supplementary Figure (Fig. R10).

In Fig. 3, the emission intensity versus the depletion intensity of NaYF₄:Nd/Yb/Er and Nd/Yb/Ho nanoparticles were combined into one figure (Fig. R6c). We presented the emission inhibition full spectra of Pr-, Eu-, Tm- and Tb-nanoparticles in main article (Fig. R6e-h) and combined their energy level diagram into supplementary Figure to remove the repeated information (Fig. R7).

Changes made in the revised manuscript:

Please see Figures R2, R6, R9 and R10 (Figures 3 and 4 in the revised manuscript, supplementary Figures 1 and 7 in the revised Supplementary Information).

Figure R9 (supplementary Figure 1 in the revised manuscript) Simulated population inversion between the $4F_{3/2}$ and $4I_{11/2}$ states for different decay lifetimes. a Classic four-level system in Nd^{3+} offers a large emission cross-section (σ), a long energy level lifetime (τ) and a long depletion wavelength (λ), allowing high-efficiency stimulated emission depletion featuring low saturation intensity, I_{sat} . b Population inversion for different decay lifetimes of $4I_{11/2}$, with the $4F_{3/2}$ state lifetime $\tau = 200 \mu s$. c Population inversion for different decay lifetimes of $4F_{3/2}$, with the $4I_{11/2}$ state lifetime $\tau = 2 \mu s$. d Simulated depletion saturation intensities (I_{sat}) of the one-photon emission from the $4F_{3/2}$ state, with different population inversion ratio between $4F_{3/2}$ and $4I_{11/2}$, showing significant decrease with the increasing inverted population.

Figure R10 (supplementary Figure 7 in the revised manuscript). **a** The depletion efficiencies of 1-, 2-, and 3-photon emissions are 50%, 75% and 87.5% with 50% depletion of sensitizer. **b** The Energy diagram of cascade amplified depletion modelling in the STExD process. **c** The calculated depletion rates ($\eta^{-1} = 1 + I/I_{\text{sat}}$) of multiphoton emissions versus depletion intensity.

c) because of cascade nature of the process the information about kinetics of the energy transfers and time required to achieve steady-state conditions is required to be studied and discussed. the "immediate" does not exhaust the topic.

Response: We thank the reviewer for his/her suggestion. We realize that the description “immediate” was inaccurate. In the original manuscript, we simply meant that this process happened very efficiently. As requested, we have implemented more experiments to analyze the dynamics of the rise and decay processes of the 588 nm emission band from Nd-nanoparticles under the co-irradiation of pulsed 740 nm and CW 1064 nm beam (**Fig R11**). Under the co-irradiation of CW 740 nm and pulsed 1064 nm beam, we analyzed the kinetics of the energy transfers of 588 nm emission. The time required to achieve steady-state condition varied from $51.7 \pm 1.9 \mu\text{s}$ to $13.8 \pm 0.1 \mu\text{s}$ when the intensity of the 1064 nm beam varied from 10 kW cm^{-2} to 1 MW cm^{-2} .

Changes made in the revised manuscript:

Please see Figures R11 (supplementary Figure 11 in the revised Supplementary Information).

Sentences modified (Page 4, results): “When the 1064 nm laser was applied, all emissions were largely inhibited, which can be attributed to the stimulated emission process (${}^4\text{F}_{3/2} \rightarrow {}^4\text{I}_{11/2}$) induced by the 1064 nm laser as discussed above.”

Sentences modified (Page 5, results): “To investigate the optical depletion kinetics, the time-resolved intensity curves were acquired through modulating excitation beam or depletion beam. With the co-irradiation of pulsed 740 nm beam and CW 1064 nm beam, the measured decay lifetime of the ${}^2\text{P}_{1/2}$ state decreased from 30.8 ± 0.5 to $12.7 \pm 0.2 \mu\text{s}$ when increasing the depletion intensity (**Fig. S11a**), suggesting the effect of the sensitizing ${}^4\text{F}_{3/2}$ states, greatly quenched by stimulated emission, on the decay time of this higher-lying level^{18,29,30}. At the exaction of CW 740 nm beam, the emission can be obviously inhibited by the pulsed 1064 nm beam (**Fig. S11b**). The time required to reach a new steady-state decreased from 51.7 ± 1.9 to $13.8 \pm 0.1 \mu\text{s}$ with the increasing depletion intensities.”

- 29 Peng, X. *et al.* Fast upconversion super-resolution microscopy with 10 μs per pixel dwell times. *Nanoscale* **11**, 1563-1569 (2019).
- 30 Bergstrand, J. *et al.* On the decay time of upconversion luminescence. *Nanoscale* **11**, 4959-4969 (2019).

Figure added (Figure R11, supplementary Figure 11 in the revised Supplementary Information): We added the Time-dependent kinetic processes of the ${}^2\text{P}_{1/2}$ state in $\text{NaYF}_4:\text{Nd}$ (3%) nanoparticles with different depletion intensities in the revised Supplementary Information to analyze the kinetics of the energy transfers and time required to achieve steady-state conditions.

Figure R11 (supplementary Figure 11 in the revised manuscript) Time-dependent kinetic processes of the ${}^2\text{P}_{1/2}$ state in $\text{NaYF}_4:\text{Nd}$ (3%) nanoparticles with different depletion intensities. **a** The excitation beam was modulated with chopper (700 Hz, $I_{740} = 78.3 \text{ kW cm}^{-2}$) and the intensity of CW depletion beam was controlled, $I_{1064} = 0, 0.01, 0.02, 0.05, 0.1, 0.2, 0.5$ and 1 MW cm^{-2} . The rising time decreased from 79.1 ± 0.5 to $46.5 \pm 0.7 \mu\text{s}$. The lifetime decreased from 30.8 ± 0.5 to $12.7 \pm 0.2 \mu\text{s}$. **b** The intensity of CW excitation beam was $I_{740} = 78.3 \text{ kW cm}^{-2}$, and the intensity of modulated depletion beam (700 Hz) was controlled, $I_{1064} = 0, 0.01, 0.02, 0.05, 0.1, 0.2, 0.5$ and 1 MW cm^{-2} . The time required to achieve steady state conditions decreased from 51.7 ± 1.9 to $13.8 \pm 0.1 \mu\text{s}$. Error bars are defined as the standard deviation of $n = 3$ independent measurements.

3. "we achieve emission depletion for a wide range of emitters" is actually not correct, because this is the sensitization mechanism which is 'depleted' and not the emission of wide range of emitters - it doesn't come to emitting state population in 740+1064 combined together. Moreover, the issues explained in the abstract are not fully addressed - MW/cm^2 depletion intensity, which is high anyway, is still required. Moreover, what "multi-imaging nanoscopy" means?

Response: We thank the reviewer for his/her comments.

We agree with the reviewer that in STExD process, the direct depletion happens to the sensitization via stimulated emission and the emissions of many activators was indirectly inhibited. In this revision, we adopted the term "emission inhibition" to describe the STExD process accurately.

We agree with the reviewer that in this work the depletion intensity at the order of MW/cm^2 is

not very low. But it has been reduced by two or three orders of magnitudes compared to that of the traditional STED techniques. STED is a powerful super-resolution microscopy technique. But, one central problem remains largely unsolved: how to achieve STED super-resolution at low beam intensities? The STED microscopy is principally driven by a high photon flux (typically in the range of 0.1-10 GW/cm², ultrafast lasers) to achieve the maximum depletion efficiency. The cascaded depletion amplification effect of STExD provides a new, effective approach for reducing the required laser intensity. In addition, ultrafast lasers can be abandoned in our study and STExD enables low-power, CW, near infrared lasers for STED super-resolution implementation. Accordingly, we also modified the words in the revised Abstract.

"Multi-imaging nanoscopy" means that our STExD setup can achieve super-resolution imaging of multiple, different luminescent materials without altering the setup configurations. As suggested, this sentence has been changed in revised manuscript to make it clearer.

Changes made in the revised manuscript:

Please see the revision of term "emission inhibition" in the revised manuscript and Supplementary Information.

Sentences modified (Page 1, abstract): "With NaYF₄:Nd nanoparticles, we demonstrate an ultrahigh depletion efficiency of $99.3 \pm 0.3\%$ for the 450 nm emission with a ~~ultra-low~~ saturation intensity of 23.8 ± 0.4 kW cm⁻²."

Sentences modified (Page 1, abstract): "The strategy expounded here promotes single wavelength-pair nanoscopy for multi-chromatic probes and for multi-color imaging under low intensity level NIR-II CW laser depletion."

Sentences modified (Page 2, introduction): "In addition, high-precision overlapping of multiple pairs of lasers is also typically required for multi-chromatic probes and for multi-color super-resolution imaging to match their different energy states, which results in high complexity, high cost and low usability^{13,14} (Fig. 1a)."

4. The mechanism is clear for me and any specialist in Ln³⁺ spectroscopy, but for more general public, more details (fig.1) will definitely be beneficial for the readers.

Response: We thank the reviewer for his/her suggestion. We have adjusted the Fig. 1 and removed the balls that may cause confusion (Fig. R12). We have also modified the sentences in the Introduction section about the status of STED microscopy.

Changes made in the revised manuscript:

Please see Figures R12 (Figure 1 in the revised manuscript).

Sentences modified (Page 2, introduction): "In addition, high-precision overlapping of multiple pairs of lasers is also typically required for multi-chromatic probes and for multi-color super-resolution imaging to match their different energy states, which results in high complexity, high cost and low usability^{13,14} (Fig. 1a). Meanwhile, the spectral overlap among the utilized dyes (typically with broad emission spectra) would also lead to cross-talk between channels¹⁵, and thereby compromises its utility and impact (Fig. 1a)."

Figure R12 (Figure 1 in the revised manuscript) Schematics of two distinct emission inhibition mechanisms. **a** In traditional STED, the wavelengths of the excitation and depletion laser beams typically have to be carefully adapted with the varied spectroscopic properties of multi-chromatic probes. Multiple pairs of laser beams have to be temporally and spatially overlapped with high-precision for multi-color imaging. For multi-chromatic imaging, a unified excitation or depletion light wavelength would lead to low excitation or depletion efficiency and possible channel cross-talk. **b** The proposed stimulated-emission induced excitation depletion (STExD) strategy can generate depopulation **simultaneously** for multiple emitting states of multi-chromatic probes **by utilizing** a single depletion wavelength to **deexcite** their common sensitizer. In STExD, the depletion laser is defined by the common sensitizer, and thereby the strong dependence of the depletion wavelength on the emitting states and emission colors will be abnormally broken.

5. there is no such thing as "long living transition"

Response: We thank the reviewer for his/her comment. We have removed this description and only retained the term of "long-lived energy level".

Changes made in the revised manuscript:

Sentences modified (Page 3, results): "To implement this, we employ Nd^{3+} as the sensitizer²² as it has a typical quasi-four-level configuration suitable for lasing²³. As a long-lived energy level, the $^4\text{F}_{3/2}$ state of Nd^{3+} is in stark contrast with the ultra-fast decaying $^4\text{I}_{11/2}$ state."

6. "We also found that the optical depletion of the 655 nm emission from the $^4\text{F}_{9/2}$ state is more efficient than that of the 540 nm emission from the $^4\text{S}_{3/2}$ state, which was attributed to depletion-beam-driven excited state absorption (ESA, $^4\text{F}_{9/2} \rightarrow ^2\text{H}_{9/2}$)." indeed, the depletion efficiency

should be the same for 540 and 650 nm emission if this is the same mechanism of Nd sensitization undergoing depletion. But I am not sure I understand the explanation on ESA - if ESA occurs, the $^4F_{9/2} \rightarrow ^2H_{9/2}$ emission should be less prone to depletion, isn't it?

Response: We thank the reviewer for his/her comments. In Nd/Yb/Er system, the observed depletion efficiencies for 540 and 650 nm emission were different, and we think that the ESA process of 1064 nm caused the result (**Fig. R6a, b**). During the process of emission inhibition, the 655 nm emission was partially inhibited by the 1064 nm beam via the ESA process of $^4F_{9/2} \rightarrow ^2H_{9/2}$, which can in turn contribute to the 525 nm and 540 nm emission and therefore slightly weaken the overall emission inhibition of green emissions (the emissions from the states: $^4F_{9/2} \rightarrow ^2H_{9/2}$, as the reviewer mentioned, are less prone to depletion). In this revision, we further measured Er³⁺-doped nanoparticles under 800 nm and 1064 nm co-excitation. The 655 nm emission was inhibited by the 1064 nm beam, and the brightness of 525 nm and 540 nm emission bands are enhanced. The giant enhancement of 472 nm and 506 nm emission bands can be attributed to the ESA process of $^4S_{3/2} \rightarrow ^2K_{15/2}$ (**Fig. R13**).

Figure R13. a Emission spectra of NaYF₄:Er (5%) nanoparticles under the co-irradiation of 800 nm and 1064 nm ($I_{800} = 96.8 \text{ kW cm}^{-2}$; $I_{1064} = 2.5 \text{ MW cm}^{-2}$). **b** Energy diagram illustrating the ESA processes driven by the 1064 nm beam.

7. "Note that this STExD method can also circumvent the constraints of the emerging up-conversion super-resolution techniques^{17,18}" - which are? Above that, I also think the authors should be more modest in their conclusions about their work again state of the art, and also discuss the drawbacks of their work such as poor signal to noise or background ratio (which is not discussed at all), large size (as compared to organic dyes) of the luminescent probes which get even larger when trying to passivate the surface of NPs, the dispersion between excitation and depletion may still cause issues with the two beams overlap, as well as fair comparison with other super-resolution techniques such as stochastic methods etc.

Response: We thank the reviewer for his/her comments.

As the reviewer mentioned, the brightness of single nanoparticles was relatively weak in the current STExD work. The particle sizes (sub-10 nm) are larger than that of organic dyes, but

comparable to antibodies used for immunolabeling. We agreed that new material design and synthesis need to be explored to make the nanoparticles smaller and brighter, such as by the surface coordination of organic molecules which greatly enhanced the brightness with no need to increase the size of nanoparticles (*Nat. Photonics* **2021**, **15**, 732). Regarding the signal-to-noise ratio issue, it is worth noting that in many studies, the photon number of a single molecule or a single nanoparticle is also not that large, for example, the photon number of single upconversion nanoparticle mentioned in ref (*Nature* **2017**, **543**, 229. Fig. 1b) is about 250 counts per 4 ms, and the photon number of rsEGFP labelled on vimentin mentioned (*Nat. Photon.* **2016**, **10**, 122. Fig. 4b) is about 15 counts per 350 μ s. The photon number of our nanoparticles is comparable to these. In addition, the performance of the detector and the optical setup will directly affect the signal to noise and background ratio. Further optimizations in terms of nanoparticles engineering and optical systems, e.g., enhancing particle brightness or using a high-performance detector, can probably further improve the signal-to-noise ratio of imaging. In the revised manuscript, we slightly adjusted the image color and the image contrast in the Figure 4.

The "constraints" means that the up-conversion STED is only achieved in Yb³⁺-Tm³⁺ system (*Nature* **2017**, **543**, 229; *Nat. Commun.* **2017**, **8**, 1058). In addition, the excitation and emission modes of microscopy are all fixed, and the depletion light needs to act on the emission energy states directly to depopulation, which is essentially similar to traditional STED of organic dyes and quantum dots. In contrast, the established simple and versatile STExD strategy in this work provides a universal approach to generate reversible photo-switchable upconversion nanoparticles for a broad spectrum of lanthanide emitters. Using the strategy of terminating the same energy supply for different, alternative emitters, we successfully achieved emission depletion for many different emitters (e.g., Nd³⁺, Yb³⁺, Er³⁺, Ho³⁺, Tm³⁺, Pr³⁺, Gd³⁺, Tb³⁺ and Eu³⁺) using the same laser-pair (see Figure 3 in the revised manuscript).

We agree with the reviewer that it would be better if the wavelengths for two laser beams are closer or the same. But actually, the issue of chromatic aberration in most of STED systems does not matter that much. For many commercially objective lenses available, the achromatic technology has been quite mature, which can help focus and overlap the laser beams. In addition, we also added two pair of lenses to adjust the divergence of two laser beams separately and can adjust their foci to the same plane. The precise 3D overlap of two CW beams can be achieved (similar to widely used STED systems).

We removed the table of technical comparison which may cause debates (**supplementary Table 6** in the original Supplementary Information).

Changes made in the revised manuscript:

Please see Figure 4 in the revised manuscript).

Sentences added (Page 2, introduction): “However, photo-switching in lanthanide nanoparticles is unfortunately rather limited and difficult to tune, and the depletion laser needs to directly act on the emitting activators for emissive-state depopulation, which is essentially the same as in

traditional STED implemented with organic dyes and quantum dots^{17,18}.”

Sentences modified (Page 9, discussion): “This STExD mechanism can also circumvent the constraints of very limited photo-switching upconversion activators for the emerging upconversion STED super-resolution techniques^{17,18}.”

Sentences added (Page 8, discussion): “The continuous development of new design and synthesis methods is expected to produce even smaller and brighter nanoparticles, which will further improve the detection signal-to-noise ratio, imaging resolution and the immunolabeling effect for diverse applications. The imaging speed can also be further increased by employing multi-focus or spinning disk scanning modes³³.”

33 Schueder, F. *et al.* Multiplexed 3D super-resolution imaging of whole cells using spinning disk confocal microscopy and DNA-PAINT. *Nat. Commun.* **8**, 2090 (2017).

8. Some typos should be carefully checked, e.g. "depletionn" etc. The article is also not equally good in terms of clarity/focus/level of details along its whole length.

Response: We thank the reviewer for the comment. We checked and corrected the typos throughout the manuscript. In addition, we reorganized Figs. 1-3 and contents to discuss the mechanism in detail for general public. We presented full-spectra emission inhibition of all Nd-sensitizing nanoparticles in main article and measured the depletion efficiencies of different emission bands of all nanoparticles. We added new experiments to build trust about reliability of analysis. We analyzed the effect of doping concentration and excitation wavelength. We measured the kinetics of the energy transfers and time required to achieve steady-state conditions of metastable energy level of Nd³⁺. We analyzed the effect of ESA processes on excitation and emission inhibition of different Nd-sensitizing systems. With modification, the core mechanism of energy transfer excitation and inhibition about the STExD strategy was highlighted and analyzed with more details. We believe that the clarity/focus/level of details have been significantly improved. In addition, this work involves nanoparticle engineering, a lot of spectroscopic studies, and optical super-resolution microscopy and biological applications, and it also includes in-depth systematic theoretical calculations, which makes this article really not that short. However, in the process of writing, we also paid attention to the length of the manuscript. The length of the current manuscript is comparable to most of the articles published in *Nature Communications*.

In closing, we would like to thank the reviewer again for taking the time to review our manuscript. We hope that the reviewer will find this revised manuscript substantially improved.

REVIEWERS' COMMENTS

Reviewer #1 (Remarks to the Author):

The authors have answered the remarks made by the reviewers, added experiments and substantially altered their manuscript. On my side, I am satisfied with the clarifications they describe and with their modification of the text/figures/tables, both in the main text and in the SI.

Despite some still-remaining problems with the technology proposed (e.g. low brightness), it is my opinion that this work is a real step forward towards better nanoscopy systems and will stimulate further research.

Therefore I consider it is ready for publication in Nat. Commun.

Reviewer #2 (Remarks to the Author):

I am satisfied with the response the authors prepared for extensive comments from me and the other reviewer. I think these are very nice studies and a very nice concept, and now I am ready to support this work and suggest it can be accepted.

Title: Achieving low-power, single-wavelength-pair nanoscopy with NIR-II CW laser for multi-chromatic probes

Point-by-Point Response to Reviewers

Referee #1 (Remarks to the Author):

The authors have answered the remarks made by the reviewers, added experiments and substantially altered their manuscript. On my side, I am satisfied with the clarifications they describe and with their modification of the text/figures/tables, both in the main text and in the SI.

Despite some still-remaining problems with the technology proposed (*e.g.*, low brightness), it is my opinion that this work is a real step forward towards better nanoscopy systems and will stimulate further research.

Therefore, I consider it is ready for publication in *Nat. Commun.*

Response: We would like to express our sincere appreciation to the reviewer for his/her effort to review this manuscript and his/her very positive evaluation and recommendation on our work.

Reviewer #2 (Remarks to the Author):

I am satisfied with the response the authors prepared for extensive comments from me and the other reviewer. I think these are very nice studies and a very nice concept, and now I am ready to support this work and suggest it can be accepted.

Response: We would like to express our sincere appreciation to the reviewer for his/her effort to review this manuscript and his/her very positive evaluation and recommendation on our work.